

# MinSIA v1: a lightweight and efficient implementation of the shallow ice approximation

Stefan Hergarten[1]

[1]Institut für Geo- und Umweltnaturwissenschaften, Albert-Ludwigs-Universität Freiburg, Albertstr. 23B, 79104 Freiburg, Germany

**Correspondence:** Stefan Hergarten
(stefan.hergarten@geologie.uni-freiburg.de)

**Abstract.** Simulations of ice flow have recently been boosted to an unprecedented numerical performance by machine learning techniques. This paper aims at keeping classical numerics competitive in this field. It introduces a new numerical scheme for the shallow ice approximation. Key features are a semi-implicit time-stepping scheme in combination with a dynamic smoothing of the nonlinearity in the slope-dependence of the flow velocity. As a first step, the software MinSIA presented here provides a lightweight implementation of the new scheme in MATLAB. An implementation in Python is under development. MinSIA is designed for simulations with several million nodes on standard desktop PCs and allows for spatial resolutions of 25 m or even finer. The numerical scheme performs particularly well for heavily glaciated topographies with moderately inclined ice surfaces. In turn, the advantage of the scheme decreases slightly for alpine topographies with steep walls during phases of moderate glaciation.

## 1 Introduction

Numerical glacier and ice-sheet models have been widely used in the context of climate change (e.g., Goelzer et al., 2020; Seroussi et al., 2020), paleoglacial reconstruction (e.g., Seguinot et al., 2018; Weber et al., 2021), and long-term landform evolution (e.g., Braun et al., 1999; Egholm et al., 2012). While models of ice dynamics typically include a multitude of processes beyond the flow of ice (e.g., meltwater flow and heat transport), the latter is typically by far the computationally most expensive part.

Three-dimensional simulations of the Stokes equations with a free surface (as implemented, e.g., in the model Elmer/Ice) and the shallow ice approximation (SIA) are end-members in the hierarchy of ice-flow models. The SIA considers vertically averaged velocities, assumes hydrostatic pressure, and neglects all stresses arising from horizontal shearing. In between, there are two-dimensional approaches that account for these stresses, such as the second-order approximation iSOSIA proposed by Egholm et al. (2011) and the Parallel Ice Sheet Model (PISM), which includes the shallow shelf approximation (Bueler and Brown, 2009).

The numerical expense of all these models depends strongly on spatial resolution. Seguinot et al. (2018) achieved a spatial resolution of 1 km in a simulation covering the entire Alps ($900 \times 600$ km) over a time span of 120,000 yr. However, these



simulations required high-performance computing facilities and a resolution of 1 km is by far not perfect for a complex
topography with many valleys of various sizes.

The still limited performance of two-dimensional models arises from a combination of assuming hydrostatic pressure and the
explicit time-stepping scheme implemented in almost all contemporary models (Bueler, 2023). The main problem is that local
variations in the slope of the ice surface generate a horizontal pressure gradient in the ice column down to the bed and thus affect
the velocity of the entire ice column. As a consequence, all depth-averaged models, including the SIA in its simplest form, are
susceptible to short-wavelength oscillations in surface elevation. Explicit schemes can only avoid the resulting instability by
small time increments, which decrease quadratically with decreasing grid spacing.

The limited progress concerning the numerical efficiency has motivated the development of alternative approaches. In the
context of large-scale landform evolution, Deal and Prasicek (2021) and Hergarten (2021) independently developed an ap-
proach to simulate erosion of valley glaciers without simulating the flow of ice explicitly. Implemented in the landform evo-
lution model OpenLEM, it allows for simulations with spatial resolutions of less than 100 m over millions of years, and a
benchmarking with the iSOSIA model yielded promising results (Liebl et al., 2023). However, this approach circumvents the
numerical issues of simulating the flow of ice for a specific application, but does not solve them.

The most remarkable alternative approach is based on deep learning. The Instructed Glacier Model (IGM) (Jouvet et al.,
2021) is an ice flow emulator that uses a Convolutional Neural Network trained by a two- or three-dimensional fluid-dynamical
model. It achieves a speedup of up to three orders of magnitude compared to the original models. Recently, a version has been
developed that even does not require a numerical model for training, but integrates a Convolutional Neural Network directly in
the numerical solver (Jouvet and Cordonnier, 2023).

The purpose of this paper is to challenge the hypothesis that the potential for further improvements in computational effi-
ciency with classical numerical methods is limited (Jouvet et al., 2021). The approach presented in the following combines a
semi-implicit time-stepping scheme as already implemented in a few models (Bueler, 2016; Wirbel et al., 2018) with a novel
dynamic stabilization against small-wavelength oscillations.

## 2  Scope and governing equations

MinSIA simulates the flow of ice on a given topography $b(x, y)$ in a two-dimensional approximation in Cartesian coordinates.
The vertical ice thickness $h(x, y, t)$ is the considered variable. The rate of change in ice thickness is defined by the volumetric
balance equation

$$\frac{\partial h}{\partial t} = -\text{div}\boldsymbol{q} + r \tag{1}$$

where $\boldsymbol{q}(x, y, t)$ is the two-dimensional flux density (volume per time and cross-section length), $r(x, y, t)$ the net rate of ice
production and melting (thickness per time), and div the two-dimensional divergence operator.

Combining efficiency with maximum flexibility was the design goal of MinSIA. At the lowest level of complexity, simple
predefined functions for $\boldsymbol{q}$ and $r$ are offered, which do not require much additional programming for running the model. These





functions can be combined with spatially and temporally variable parameter values for more complex applications. Finally, more or less completely custom functions can be included, which also allow for coupling MinSIA with other models., e.g., for temperature.

The simple model for the flux density is based on a decomposition into ice deformation and sliding along the bed,

$$\boldsymbol{q} = \boldsymbol{q}_{\mathrm{d}} + \boldsymbol{q}_{\mathrm{s}}. \tag{2}$$

The contribution of deformation is described by the relation

$$\boldsymbol{q}_{\mathrm{d}} = -f_{\mathrm{d}} h^5 (\cos\beta)^8 |\nabla s|^2 \nabla s, \tag{3}$$

where

$$s(x, y, t) = b(x, y) + h(x, y, t) \tag{4}$$

is the elevation of the ice surface and $\nabla s$ is its gradient. This relation is based on Glen's flow law, which assumes that the strain rate is proportional to $\tau^n$ with $n = 3$, where $\tau$ is the shear stress. For a detailed discussion of this relation and the value of the exponent $n$, readers are referred to Cuffey and Paterson (2010). The factor $f_{\mathrm{d}}$ mainly depends on temperature in reality.

The factor involving the cosine of the slope angle $\beta$ of the ice surface with

$$\cos^2\beta = \frac{1}{1 + \tan^2\beta} = \frac{1}{1 + |\nabla s|^2} \tag{5}$$

is a simple correction for steep slopes. It takes into account that the thickness perpendicular to the surface is lower than the vertical thickness and that driving forces by gravity are proportional to $\sin\beta$ instead of $\tan\beta = |\nabla s|$. The correction is explained in the derivation of Eq. (3) in Appendix A. It should, however, be mentioned that it is less elaborate than the correction for rapid mass movements developed by Savage and Hutter (1989) and only exact if the ice surface is parallel to the bed. In particular, it does not capture the effect of steep walls in a valley. Therefore, it is only included as an option.

For sliding at the bed, the relation

$$\boldsymbol{q}_{\mathrm{s}} = -f_{\mathrm{s}} h^3 (\cos\beta)^5 |\nabla s|^2 \nabla s, \tag{6}$$

is used. It combines the ideas of Weertman (1957) and Bindschadler (1983) (see also Cuffey and Paterson, 2010, chapter 7). As a strong simplification, Eq. (6) relies on the assumption that the water pressure at the bed is proportional to the normal stress, following some simple models of glacial erosion that do not consider meltwater hydraulics explicitly (Braun et al., 1999; Deal and Prasicek, 2021; Hergarten, 2021). A short derivation of Eq. (6) is given in Appendix B.

It should, however, be emphasized that modeling sliding along the bed is still one of the major challenges in this field. It is even questionable whether this process can be described well by the SIA or whether lateral stress components have to be taken into account.

To make the model more flexible, the values of $f_{\mathrm{d}}$ and $f_{\mathrm{s}}$ may vary in space and time. This would, e.g., allow for taking into account their dependence on temperature in combination with a thermal model.



The generic form of the flow law,

$$\boldsymbol{q} = -D(h, |\nabla s|)\nabla s, \tag{7}$$

with a user-defined function $D(h, |\nabla s|)$, offers the highest degree of flexibility. Formally, it looks like a nonlinear diffusion equation, although it is mathematically not a diffusion equation due to the occurrence of $\nabla s$ in $D$. Nevertheless, $D$ is denoted diffusivity in the following.

Taking into account that $\frac{\partial h}{\partial t} = \frac{\partial s}{\partial t}$, the considered differential then reads

$$\frac{\partial s}{\partial t} = \mathrm{div}\left(D(h, |\nabla s|)\nabla s\right) + r, \tag{8}$$

where the specific version defined by Eqs. (3) and (6) assumes

$$D(h, |\nabla s|) = \left(f_{\mathrm{d}}h^5(\cos\beta)^8 + f_{\mathrm{s}}h^3(\cos\beta)^5\right)|\nabla s|^2. \tag{9}$$

Different approaches are also included for the net rate of ice production and melting, $r(x, y, t)$. The simplest model is piecewise linear with an equilibrium line altitude (ELA) $z_{\mathrm{e}}$ and a maximum rate $r_{\mathrm{max}}$,

$$r(z) = \begin{cases} \min\left(g_+\left(z - z_{\mathrm{e}}\right), r_{\mathrm{max}}\right) & z \geq z_{\mathrm{e}} \\ g_-\left(z - z_{\mathrm{e}}\right) & z < z_{\mathrm{e}} \end{cases} \quad \text{for} \tag{10}$$

applied to the ice surface ($z = s(x, y, t)$), where $g_+$ is the accumulation gradient (net rate of ice production per meter above the ELA) and $g_-$ is the ablation gradient (net rate of melting per meter below the ELA). A typical application would be a time-dependent ELA while keeping the other parameters constant. However, MinSIA allows for a variation of all parameter values in space and time as well as for replacing Eq. (10) by a custom function. Alternatively, $r(x, y, t)$ can also be defined directly for each cell of the grid at each time.

## 3 Numerical scheme and implementation

### 3.1 Semi-implicit time stepping

MinSIA employs a semi-implicit time-stepping scheme. This scheme combines an explicit discretization of the diffusivity with an implicit discretization for the diffusion process itself. The respective time-discrete version of Eq. (8) reads

$$\frac{s(t + \delta t) - s(t)}{\delta t} = \mathrm{div}\left(D(t)\nabla s(t + \delta t)\right) + r(t). \tag{11}$$

This semi-implicit scheme already ensures stability for arbitrary time increments $\delta t$. However, there are some caveats that require a specific treatment in order to achieve a reasonable accuracy.





## 3.2 Finite volumes with upstream diffusivity

MinSIA uses a finite-volume discretization of the domain with square cells. Since the finite-volume approach is based on the flux densities across the boundaries between adjacent cells, it requires the diffusivity $D(h, |\nabla s|)$ at the cell boundaries. MinSIA uses an upstream scheme here, which means that the diffusivity of the node with the higher ice surface is adopted for the respective cell boundary.

The upstream scheme helps to maintain the volumetric balance. While the finite-volume scheme itself preserves the ice volume exactly, the semi-implicit scheme does not see the obvious condition $h \geq 0$ (equivalent to $s \geq b$) since $D$ is fixed during the time step. Therefore, negative values of $h$ may occur, which have to be compensated afterwards. Setting negative values to zero causes a systematic increase in total volume. This problem predominantly occurs at the contact of valley glaciers with ice-free slopes. If $D > 0$ at the contact of a glacier cell and a (higher) slope cell, the slope cell will deliver non-existing ice to the glacier. The upstream scheme ensures $D = 0$ in this situation and thus avoids a permanent systematic increase in ice volume. However, the upstream scheme cannot avoid that cells flip from positive to negative values of $h$ occasionally. This effect will be investigated numerically in Sect. 4.3.

## 3.3 Smoothing the slope of the ice surface

While the semi-implicit scheme is unconditionally stable for each individual time step, the explicit treatment of $D(h, |\nabla s|)$ still causes oscillations. These oscillations occur preferably at large thickness in combination with moderate slopes, where small changes in slope have a strong effect on $D$. It will be shown in Sect. 4.1 that these oscillations result in staircase-shaped ice surfaces and may strongly affect the accuracy.

Therefore, a dynamic smoothing of the slope of the ice surface in the diffusivity $D(h, |\nabla s|)$ is the central novel idea in MinSIA. The idea was originally motivated by Airy's linear theory of ocean waves (gravity waves in an inviscid fluid with a free surface). According to this theory, oscillations in pressure and particle velocity decrease exponentially with depth below the surface. The respective depth of penetration is proportional to the wavelength. Based on this result, MinSIA uses a smoothed version of $|\nabla s|$ with the length scale of smoothing being proportional to the ice thickness $h$.

As a starting point, the slope of the ice surface is computed at each node $(i, j)$ according to

$$|\nabla s|_{i,j} = \sqrt{\left|\frac{\partial s}{\partial x}\right|_{i,j}^2 + \left|\frac{\partial s}{\partial y}\right|_{i,j}^2} \tag{12}$$

with the individual components approximated by

$$\left|\frac{\partial s}{\partial x}\right|_{i,j} = \frac{1}{2}\left(\left|\frac{s_{i,j+1} - s_{i,j}}{\delta x}\right| + \left|\frac{s_{i,j} - s_{i,j-1}}{\delta x}\right|\right) \tag{13}$$

$$\left|\frac{\partial s}{\partial y}\right|_{i,j} = \frac{1}{2}\left(\left|\frac{s_{i+1,j} - s_{i,j}}{\delta y}\right| + \left|\frac{s_{i,j} - s_{i-1,j}}{\delta y}\right|\right) \tag{14}$$

and $\delta x = \delta y$. In contrast to the usual central difference quotient, the absolute values are taken individually for the left-hand and right-hand parts. While this modification has no effect for smooth inclined ice surfaces, it avoids the formation of unrealistic spikes in the ice surface at local maxima during the simulation.





Smoothing is implemented by computing the thickness-weighted average of $|\nabla s|$ over a square,

$$\left.|\nabla s|\right|_{i,j}^{\text{smoothed}} = \frac{\sum_{k=i-m}^{i+m} \sum_{l=j-m}^{j+m} h_{k,l} \left.|\nabla s|\right|_{k,l}}{\sum_{k=i-m}^{i+m} \sum_{l=j-m}^{j+m} h_{k,l}}. \tag{15}$$

The sum extends over a square of $2m + 1$ cells along each coordinate axis. The thickness-weighted average avoids artifacts at the boundaries of ice-covered areas. For a glacier in a valley with steep walls, a simple arithmetic mean would result in an increase in smoothed slope towards the boundary if the walls are much steeper than the surface of the glacier.

To achieve a smoothing scale proportional to the thickness, a nondimensional smoothing factor $f$ is introduced in such a way that averaging is performed over a length of $2fh$ along each axis (so $fh$ in each direction). In the context of Eq. (15), this implies

$$(2m + 1)\,\delta x = 2fh \tag{16}$$

and thus

$$m = f\frac{h}{\delta x} - \frac{1}{2}. \tag{17}$$

However, $m$ will typically not be an integer number. Therefore, the weighted average is computed for the two integer numbers around $m$ and linear interpolation between the results is used.

The weighted average can be implemented efficiently by computing the cumulative sums

$$G_{i,j} = \sum_{k=1}^{i} \sum_{l=1}^{j} h_{k,l} \left.|\nabla s|\right|_{k,l} \tag{18}$$

and

$$H_{i,j} = \sum_{k=1}^{i} \sum_{l=1}^{j} h_{k,l} \tag{19}$$

at first. Then the weighted average (Eq. 15) is obtained as

$$\left.|\nabla s|\right|_{i,j}^{\text{smoothed}} = \frac{G_{i+m,j+m} - G_{i+m,j-m-1} + G_{i-m-1,j+m} - G_{i-m-1,j-m-1}}{H_{i+m,j+m} - H_{i+m,j-m-1} + H_{i-m-1,j+m} - H_{i-m-1,j-m-1}} \tag{20}$$

## 3.4 The linear solver

The linear equation system to be solved in each time step is positive definite and sparse. To improve efficiency for partly ice-covered surfaces, it is reduced to the relevant nodes (ice-covered nodes and their direct neighbors) in each step.

The preconditioned conjugate gradients (PCG) method is used for solving the linear equation system. An incomplete Cholesky factorization is used as a preconditioner, whereby the version that compensates dropped nondiagonal elements at the respective diagonal elements turned out to be particularly suitable. A direct solver is also implemented for testing.





## 4 Numerical tests

The present-day topography of the Southern Black Forest was used for the numerical tests. The considered domain consists of a $60 \times 60$ km square centered to the highest peak (Feldberg, 1493 m). The publicly available 1 m terrain model (Landesamt für Geoinformation und Landentwicklung, 2025) was downsampled to a grid spacing of $\delta x = \delta y = 25$ m, resulting in a grid of 2400 × 2400 cells.

The values $f_\mathrm{d} = 5.34 \times 10^{-5}$ m$^{-3}$yr$^{-1}$ and $f_\mathrm{s} = 3.56$ m$^{-1}$yr$^{-1}$ are assumed for the flow parameters, corresponding to the values used by Deal and Prasicek (2021). The slope correction with the $\cos\beta$ terms in Eq. (9) was active. For the net rate of ice production, the simple ELA-based approach (Eq. 10) was used with an accumulation gradient of $g_+ = 0.002$ yr$^{-1}$ and an ablation gradient of $g_- = 0.003$ yr$^{-1}$. These values are oriented on the rates assumed in the model comparison of Liebl et al. (2023). All simulations were run over a time span of 3000 yr with a constant ELA at $z_\mathrm{e} = 1000$ m, starting from an ice-free surface. This setup does not address anything that ever happened, but was designed as a simple scenario with a transition from isolated valley glaciers towards a continuous ice cap, as illustrated in Fig. 1.

Time increment was $\delta t = \frac{1}{64}$ yr and the smoothing factor was set to $f = 0.25$ as a reference. As the maximum thickness achieved at the end was $h = 637$ m, this choice corresponds to smoothing over squares of $13 \times 13$ cells (325 m; $m = 6$ in Eq. 15) at maximum. It would be desirable to use a simulation without smoothing as a reference scenario. However, this simulation already took several days on a standard desktop PC, and it will become clear in the following that a simulation without smoothing would require much smaller time increments at the high spatial resolution used here.

All simulations, except for the tests of the linear solver (Sect. 4.4), were performed either with the direct solver or with the PCG solver with a relative tolerance of $10^{-10}$. This value is small enough to ensure that the errors caused by the iterative PCG solver are negligible.

### 4.1 The effect of smoothing

As a first test, the accuracy of the solutions for different values of the smoothing factor $f$ with respect to the reference solution ($f = 0.25$, $\delta t = \frac{1}{64}$ yr) is measured. For this purpose, the empirical cumulative distribution function (CDF) of the node-by-node deviation from the reference solution is computed, derived from all nodes covered by ice in either of the two compared scenarios.

The colored curves in Fig. 2 show an increase in deviation from the reference solution ($f = 0.25$) with increasing $f$. Comparing the deviations at $t = 300$ yr (Fig. 2a) to those at $t = 3000$ yr (Fig. 2b) reveals an increase through time. This increase is, however, not only due to the accumulation of errors, but also goes along with the overall increase in ice thickness.

To get a better feeling for the accuracy, Fig. 3 shows the longitudinal valley profile defined in Fig. 1. The profile line follows the direction of the steepest descent of the bedrock topography, starting from the highest peak towards northwest. The profiles for $f = 0.5$ and $f = 1$ are visually indistinguishable from the reference profile, in agreement with the small deviations in Fig. 2. For $f = 1$, 95 % of all deviations are in the range from $-1.79$ m to $0.23$ m at $t = 3000$ yr. While the deviation in the profile



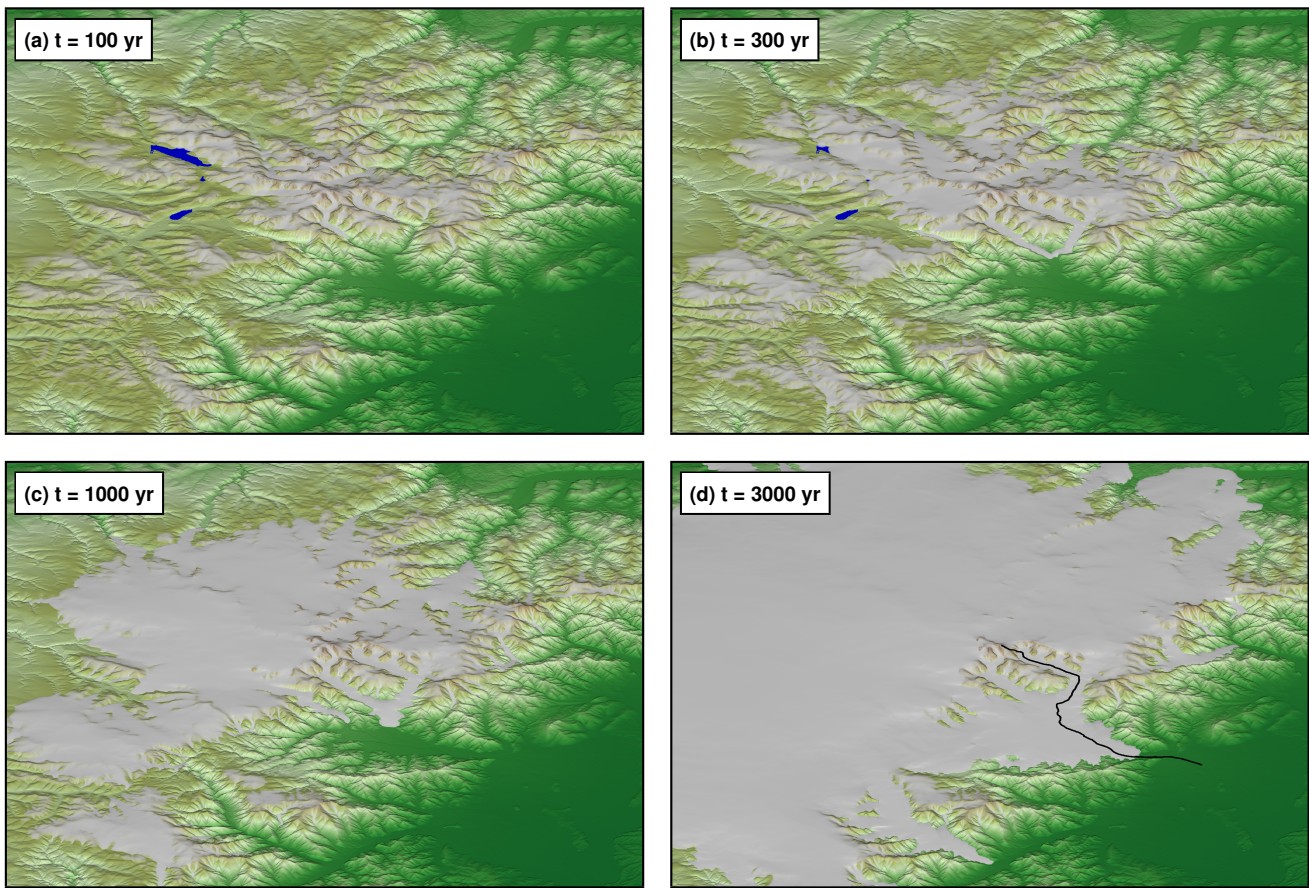

**Figure 1.** Snapshots from the reference scenario. The black line in (d) refers to the longitudinal profile analyzed in Fig. 3.

(Fig. 3) is still small for $f = 2$, the ice surface is considerably too low for $f \geq 4$, in particular in the upper part of the glacier. Here the ice thickness is almost 100 m lower for $f = 8$ than for $f = 0.25$.

The behavior is completely different for $f = 0.125$. The vast majority of all nodes has a very small deviation from the reference surface ($f = 0.25$) at $t = 300$ yr (Fig. 2a). However, the distribution already shows quite long tails at both sides, reflecting a considerable deviation for a few percent of the nodes. The deviations become much larger at $t = 3000$ yr (Fig. 2b) with a strong trend towards large elevations. This trend is also visible in the glacier profile (Fig. 3), where the ice surface is systematically too high and the glacier is 1 km too long.

Figure 3 also reveals that the strong decrease in accuracy for $f = 0.125$ goes along with staircase-shaped surface. One of the staircase-shaped sections is even considerably steeper than the reference surface. This means that the step-like shape has a negative effect on the ice flux, which must be compensated either by a greater thickness or a greater average slope.



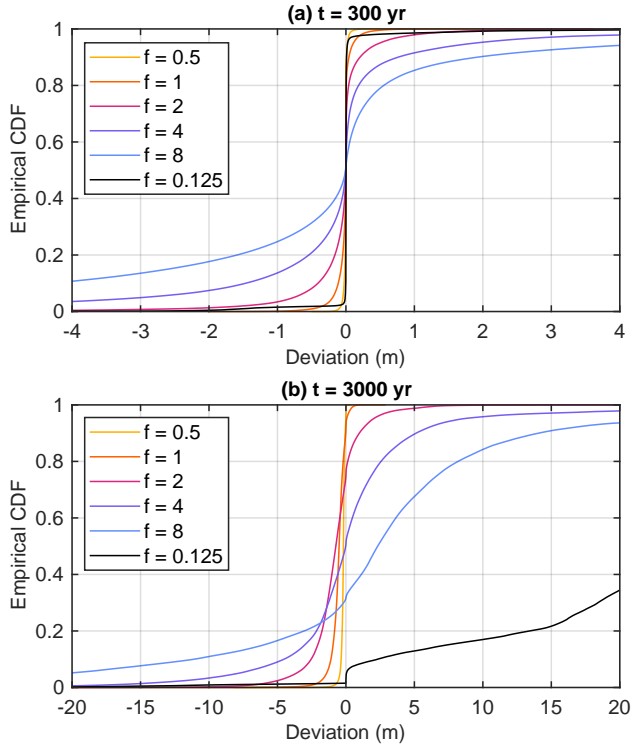

**Figure 2.** Deviation in ice thickness from the reference scenario ($f = 0.25$). The empirical CDF was derived from all nodes covered by ice in either of the two compared scenarios.

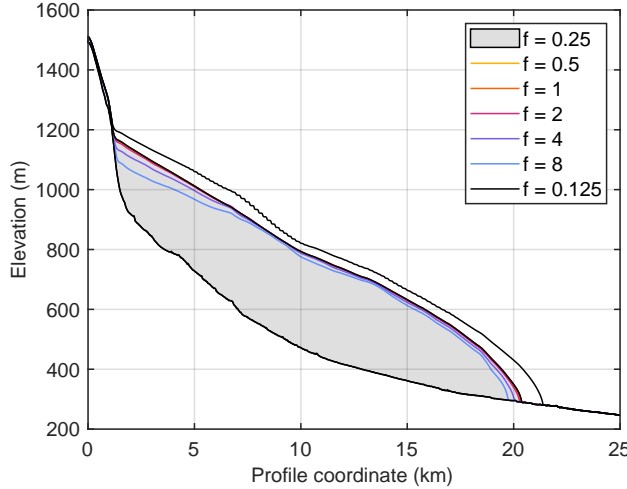

**Figure 3.** Longitudinal glacier profile along the line defined in Fig. 1 at $t = 3000$ yr for different values of $f$.





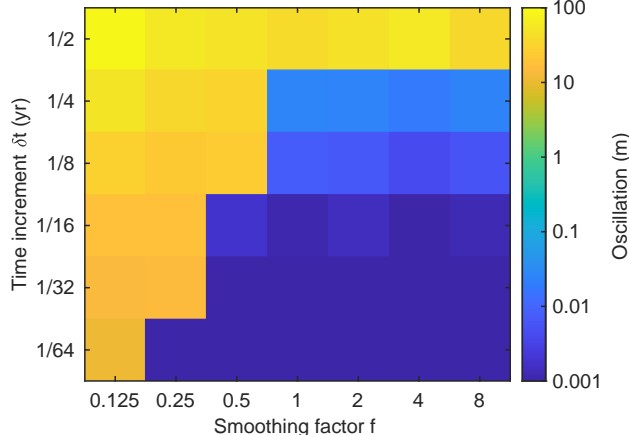

**Figure 4.** 95 % quantiles of the maximum oscillation for different values of $f$ and time increments $\delta t$.

## 4.2 The maximum time increment

Since even explicit schemes are stable for sufficiently small time increments, the occurrence of the staircase oscillation must
be related to both the smoothing factor $f$ and the time increment $\delta t$. It can be easily detected by tracking the elevation of the ice surface node by node without analyzing the topography of the ice surface. Let us define

$$
\delta h(t) = \begin{cases} h(t + \delta t) - h(t) & \\ -(h(t + \delta t) - h(t)) & \end{cases} \text{ for } \begin{array}{l} h(t) < h(t - \delta t) \\ h(t) \geq h(t - \delta t) \end{array} . \tag{21}
$$

Positive values of $\delta h$ describe points that decreased previously and increase now or increased previously and decrease now. So positive values of $\delta h$ are a measure of the oscillation. MinSIA returns the maximum value of $\delta h$ over all nodes in each time
step.

Figure 4 shows the 95 % quantiles of the oscillation (still the maximum $\delta h$ over all nodes in each step, but then the value that is not exceeded in 95 % of all time steps). Since not all changes from increasing to decreasing thickness or vice versa are necessarily numerical artifacts, the 95 % quantile is more useful than the maximum over all time steps. The exact choice of the quantile, however, is not important.

The values of the 95 % oscillations show a clear distinction with only values above $10$ m (yellow) and below $0.1$ m (blue). So there is a clearly defined (at least within a factor of two) maximum time increment for each value of $f$. There seem to be two different regimes concerning the value of $f$. While the maximum $\delta t$ increases strongly with increasing $f$ for $f \leq 1$, it seems to remain constant for $f > 1$.

At least for this example, this means that increasing the smoothing factor to values $f > 1$ does not improve performance
strongly. In combination with the good accuracy observed for $f \leq 1$ in Sect. 4.1, this finding suggests $f = 1$ tentatively as a good tradeoff between accuracy and performance.





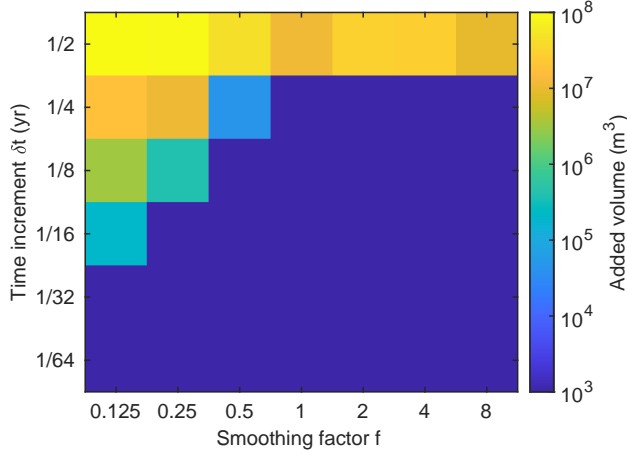

**Figure 5.** Total volume added throughout the simulation in order to keep the ice surface consistent ($h \geq 0$).

## 4.3 The volumetric balance

As discussed in Sect. 3.2, the upstream scheme cannot avoid an occasional occurrence of negative thickness values. These negative values have to be compensated (set to zero) to keep the ice surface consistent, which causes a systematic error in the volumetric balance.

Figure 5 shows the results of an analysis similar to that from Fig. 4, but for the total ice volume added throughout the simulation instead of the maximum oscillation. Effects of melting were removed from the considered volume. As an example, $2 \, \mathrm{m}$ initial thickness, $1 \, \mathrm{m}$ loss by melting, and $5 \, \mathrm{m}$ loss by outflow would be only $3 \, \mathrm{m}$ to be compensated.

The result for the added volume looks qualitative similar to the result for the oscillation (Fig. 4) with only values below $1000 \, \mathrm{m}^3$ (blue) and above $200{,}000 \, \mathrm{m}^3$ (green to yellow). The transition takes place at higher $\delta t$ than in Fig. 4 at least for $f < 1$, which means that avoiding staircase oscillations defines a lower limit for $\delta t$ than preserving the volumetric balance.

It should, however, be mentioned that even the highest added volume of about $1.4 \times 10^8 \, \mathrm{m}^3$ is small in relation to the total volume of about $5 \times 10^{11} \, \mathrm{m}^3$. To ensure that this result is not limited to advancing glaciers, two additional scenarios were considered. One of them assumes a sudden rise of the ELA to $z_\mathrm{e} = 3000 \, \mathrm{m}$ at $t = 3000 \, \mathrm{yr}$, which lets all ice vanish within 100 years. The other scenario switches off ice production and melting at $t = 3000 \, \mathrm{yr}$, so that the existing ice flows down toward the boundaries of the domain. The added volume per time is even lower than for advancing glaciers in both cases.

These results suggest that the systematic error in the volumetric balance is negligible compared to the immediate effect of the staircase oscillations on accuracy. So focus should be on limiting $\delta t$ to avoid these oscillations.





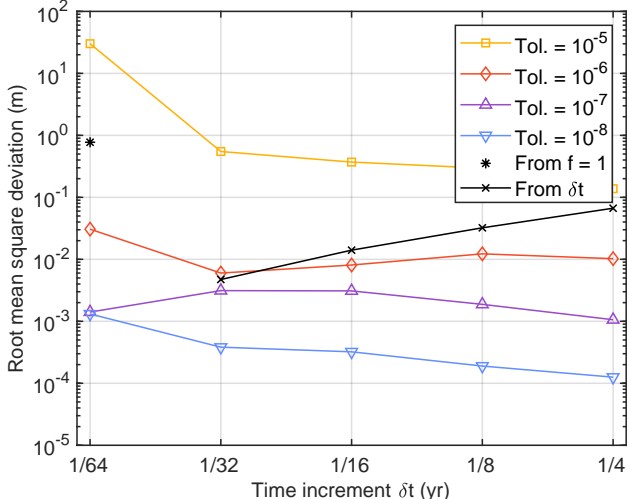

**Figure 6.** Root mean square deviations for $f = 1$ at $t = 3000$ yr. Colored lines correspond to different tolerance levels of the PCG solver. Black lines and points correspond to the error arising from smoothing ($f = 1$ vs. $f = 0.25$) and from the finite time increment (actual $\delta t$ vs. $\delta t = \frac{1}{64}$ yr).

## 4.4 The linear solver

The application of the iterative PGC solver is a tradeoff between accuracy and numerical effort. The version implemented in MinSIA uses the relative norm of the residuum (the default for the used MATLAB function) as a criterion for terminating the iteration, called tolerance in the following.

The colored lines in Fig. 6 show the error caused by the finite tolerance of the PCG solver for $f = 1$ at $t = 3000$ yr, quantified by the root mean square deviation. Following the same concept as in Fig. 2, the deviation is derived from all nodes covered by 250 ice in either of the two compared scenarios.

As long as the tolerance is not bigger than $10^{-6}$, the respective error is much smaller than the error arising from smoothing with $f = 1$. Therefore, a tolerance of $10^{-6}$ is sufficient here. However, a decrease to $10^{-7}$ increases the numerical effort by less than 10 % here and thus provides an additional safety margin at reasonable cost. These results also hold for the coarser resolutions considered in the next section.

## 255 4.5 The spatial resolution

If there is a maximum stable time increment for a numerical scheme, it typically depends on the spatial resolution. Figure 7 shows the maximum time increment that avoids staircase oscillations for grids of different resolutions. A sudden increase in maximum oscillation at a certain time increment was observed for each set of simulations and a threshold of 1 m could be used consistently.





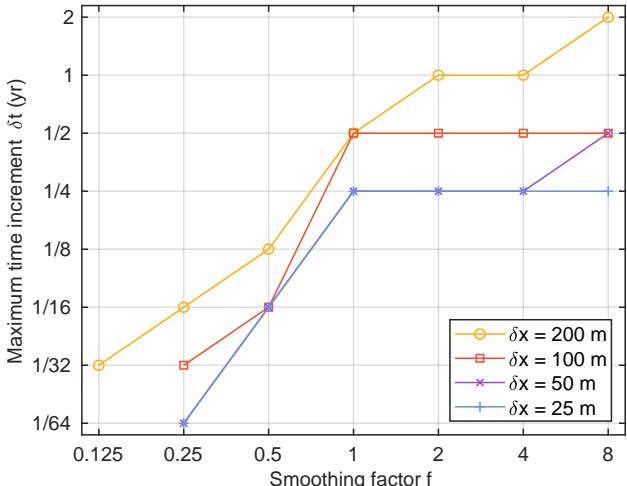

**Figure 7.** Maximum time increment that avoids staircase oscillations for different spatial resolutions.

Basically the same behavior was found for $\delta x = 25$ m, 50 m, and 100 m. In particular, the benefit of values $f > 1$ is small. Overall, a fourfold increase in $\delta x$ increases the maximum $\delta t$ by a factor of 2 at maximum. This decrease is small compared to the total number of nodes, which decreases by a factor of 16 here. So the immediate effect of the number of nodes on the computational effort is much stronger than the effect of changes in maximum $\delta t$.

Increasing grid spacing to $\delta x = 200$ m still has a minor effect on the maximum $\delta t$ around $f = 1$. However, the dependence on $f$ becomes less clear. Instead of the distinct change in behavior at $f = 1$, the maximum $\delta t$ rather increases linearly with $f$.

The simulations over the entire time span with $\delta t = 0.25$ yr took 35400 s for $\delta x = 25$ m, 6740 s for $\delta x = 50$ m, and 1290 s for $\delta x = 100$ m on an Intel Core i5-7600 CPU (3.50 GHz) from 2018. These numbers tentatively suggest that the numerical effort increases like $\delta x^{-2.4}$ at constant $\delta t$, which is not much stronger than $\delta x^{-2}$ from the number of nodes alone. Taking into account the above finding that the maximum $\delta t$ increases like $\delta x^{0.5}$ or weaker, the increase in numerical effort is still weaker than $\delta x^{-3}$.

## 5   Finding the best time increment

Finding a suitable time increment is the main challenge in setting up a simulation with MinSIA. As illustrated for the considered example in Fig. 6, the error caused by the finite time increment is quite small as long as staircase oscillations are avoided. So a time increment slightly below the limit at which staircase oscillations are initiated is typically the best tradeoff between accuracy and efficiency.

The findings from Sect. 4.5 suggest that the dependence of the maximum time increment on grid spacing is weaker than linear. Since the criteria that are typically applied to explicit schemes predict a stronger dependence, these criteria are not immediately useful for estimating the maximum time increment. In particular, the Fourier criterion for the diffusion equation



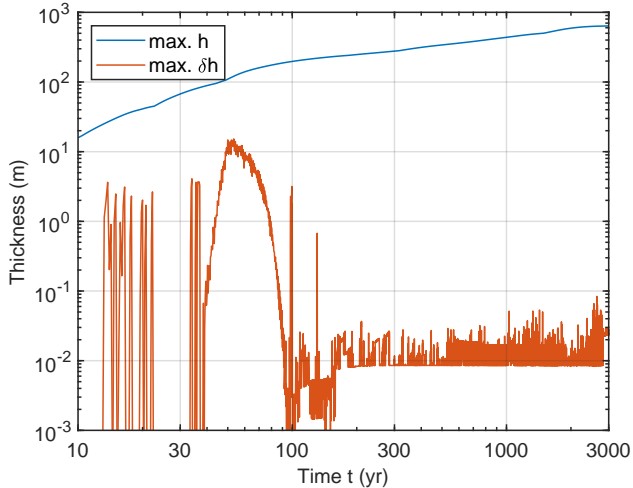

**Figure 8.** Maximum ice thickness $h$ and maximum oscillation $\delta h$ (Eq. 21) for $\delta x = 25$ m and $\delta t = 0.25$ yr.

predicts a maximum $\delta t$ proportional to $\delta x^2$, which would be a much stronger dependence than found here numerically. The
Courant–Friedrichs–Lewy (CFL) criterion for the advection equation, which roughly says that the material must not move by
more than the grid spacing in one step, predicts a maximum $\delta t$ proportional to $\delta x$, which is also is still stronger than found
here numerically. The maximum depth-averaged velocity is approximately $500\ \mathrm{m\,yr^{-1}}$ here, which means that distances of up
to $5\delta x$ are traveled per step for $\delta x = 25$ m and $\delta t = 0.25$ yr. So the CFL criterion is also exceeded absolutely.

However, the CFL criterion becomes relevant in some situations. As described in Sect. 3.3, The numerical scheme was
tailored to capture the high sensitivity of $D(h, |\nabla s|)$ to changes in $|\nabla s|$ at large ice thickness $h$. In turn, the sensitivity to
changes in $h$ is high at steep slopes. Concerning this sensitivity, the semi-implicit scheme is not much better than an explicit
scheme.

Figure 8 illustrates the weakness of the scheme at steep slopes for $\delta x = 25$ m and $\delta t = 0.25$ yr. Oscillations of several meters
occur during the early phase when the maximum thickness is still much lower than at the end. A detailed analysis of the strong
oscillations with $\delta h \geq 10$ m around $t \approx 50$ yr reveals that these affect only a few nodes. While the slope $|\nabla s|$ is greater than
$0.74$ at these nodes, the ice thickness is lower than $30$ m.

These oscillations are not a problem for the test scenario. Practically, it does not matter whether ice flows down steep slopes
uniformly or in distinct surges, provided that the oscillations do not cause large errors in the volumetric balance due to negative
values of $h$. In this example, the oscillations even cease for $t > 100$ yr due to filling up the steep slopes with ice. However, this
is not necessarily the case for other topographies.

Some additional, but preliminary tests were performed for the Rhone–Aare region in Switzerland and the Salzach region in
Austria as two examples of alpine topography. All parameter values were the same as for the Black Forest example, except for
an ELA of $z_\mathrm{e} = 1500$ m and an upper limit for the net rate of ice production (Eq. 10) of $r_\mathrm{max} = 1\ \mathrm{m\,yr^{-1}}$.





A time increment of $\delta t = \frac{1}{32}$ yr turned out to be sufficient for both scenarios at $\delta x = 50$ m, whereby the maximum ice
thickness exceeded $2000$ m in the Rhone–Aare example and $1450$ m in the Salzach example. This time increment is 8 times
smaller than that found for the Black Forest example. However, the maximum ice thickness is also more than 3 times higher
and it was already found in Sect. 4.1 that the occurrence of staircase oscillations depends on ice thickness.

However, larger time increments caused oscillations of several meters at steep slopes already in an early phase of the sim-
ulation at moderate ice thickness. These oscillations cease in the Salzach example when the valleys have been filled with ice,
which allows for increasing $\delta t$ to $\frac{1}{16}$ yr. This is, however, not the case for the Rhone–Aare example.

So the numerical efficiency is limited by the weak performance at steep slopes for moderate ice thickness in the alpine
examples. Even increasing the ELA so that the maximum ice thickness is similar to that in the Black Forest example does not
bring the maximum time increment close to the value $\delta t = \frac{1}{4}$ yr achieved there.

Since the scheme is not much better than an explicit scheme concerning the dependence of the ice flux on $h$ at steep slopes,
the CFL criterion becomes relevant here in contrast to the results from the previous section. As mentioned above, the CFL
criterion predicts a stronger (linear) dependence of the maximum $\delta t$ on $\delta x$ than found numerically for large ice thickness.
Therefore, the limitation arising from steep slopes becomes more relevant with increasing spatial resolution.

As a consequence, simulations of large alpine regions with $\delta x < 50$ m are hardly feasible on standard PCs. Beyond the
limitation of the maximum time increment, the number of iterations of the PCG scheme also contributes to the increasing
numerical effort. The results on the tolerance of the PCG solver found in Sect. 4.4 also hold for the alpine examples, whereby a
tolerance of $10^{-5}$ even seems to be slightly better than in the Black Forest example. However, the absolute number of iterations
is higher for the alpine examples. It even reaches $100$ at $h = 2000$ m, which lets the PCG solver contribute more than $80$ % to
the total effort.

However, the results obtained from the numerical tests still do not allow for estimating the maximum time increment from
the topography and the parameters directly. Finding the maximum time increment experimentally by watching the maximum
oscillation $\delta h$ (Eq. 21) seems to be the best way to estimate the maximum $\delta t$ at present.

## 6 Software description

At present, a MATLAB implementation of MinSIA is available under the GNU General Public License. A Python imple-
mentation is under development. The implementation is minimalistic and consists of a class `minsia`, which contains only a
constructor and a method `step` for performing a forward time step. Data handling and visualization are left to the user.

The constructor requires 4 mandatory arguments:

– An array `b` for the topography $b$.

– The grid spacing `dx` ($\delta x = \delta y$).

– The ice flow parameters `fd` ($f_\mathrm{d}$) and `fs` ($f_\mathrm{s}$). These parameters must be either scalar values or arrays with the same size
as `b`.




Further optional arguments are:

– The model for the net rate of ice production `prod`, either given as an array with up to 3 elements or as a function handle. A 3-element array defines the accumulation gradient $g_+$, the ablation gradient $g_-$, and the maximum rate $r_{\max}$ (in this order) for the ELA-based model (Eq. 10). A 2-element array corresponds to $r_{\max} = \infty$ and a scalar value additionally to $g_- = g_+$. If `prod` is a function handle, it must refer to a function that receives the actual elevation $s$ of the ice surface and the ELA $z_{\mathrm{e}}$ as arguments (so `prod = @(s,ela)...`) and returns the net production rate $r$. It must be able to handle array-valued arguments. If the argument `prod` is not used, the production rate must be defined in each time step by the respective argument `prod` of the method `step`.

– The scalar smoothing factor `f` ($f$). Based on the findings of Sect. 4.1, the default value is $f = 1$.

– A logical variable `slopecorr` that defines whether or not the correction for steep slopes (terms with $\cos\beta$ in Eqs. 3, 6, and 9) are used. Default is `true`.

– The relative tolerance `tol` of the PCG solver. The default value is $10^{-7}$, which is a very conservative setting according to the findings of Sect. 4.4. A value of $10^{-6}$ should not affect the accuracy strongly and should improve the performance by about 10 %. Values higher than $10^{-5}$ are not recommended. Setting the tolerance to zero switches to a direct solver.

– The diffusivity `diffus` for custom models of ice flow according to Eq. (7). It must be a handle to a function that receives the ice thickness $h$, the slope of the ice surface $|\nabla s|$, and the flow parameters $f_{\mathrm{d}}$ and $f_{\mathrm{s}}$ as arguments (so `diffus = @(h,slope,fd,fs)...`). It must be able to handle array-valued arguments.

The method `step` requires two out of the following three arguments:

– The time increment `dt` ($\delta t$) is strictly required. Since the occurrence of staircase oscillations depends on topography, ice thickness and flow parameters, it is not possible to provide a useful default value. It is instead recommended to watch the maximum oscillation and adjust $\delta t$ manually. According to the preliminary tests, it is suggested to start the search somewhere around $\delta t = 0.1$ yr.

– A scalar value or an array `ela` ($z_{\mathrm{e}}$) that defines the ELA.

– An array `prod` that defines the net rate of ice production ($r$) in each cell. If this argument is provided, the argument `ela` is not used.

The method `step` updates the thickness `h`. It returns the maximum oscillation (Eq. 21) and the ice volume that was added to compensate negative thickness values (Sect. 4.3). It also updates the class variable `slope`, which contains the smoothed slope of the ice surface, $|\nabla s|^{\mathrm{smoothed}}$ (Sect. 3.3). This property may be useful for calculating velocities. Furthermore, it updates the class variable `dh`, which contains the oscillation at each node (Eq. 21). This property may be helpful for analyzing oscillations in detail.



MinSIA assumes homogeneous Neumann boundary conditions, i.e., closed boundaries. If ice reaches the boundary, it will accumulate. Alternatively, Dirichlet boundary conditions, i.e., a given ice thickness, can be mimicked by setting the thickness to the requested value after each time step.

## 7  Summary and outlook

This paper presents a novel numerical scheme and a lightweight implementation of the SIA for the flow of ice over a given topography. Key features are a semi-implicit time-stepping scheme and a dynamic smoothing of the slope of the ice surface. The proposed scheme overcomes the limitations of explicit schemes at large ice thickness and small slopes, which are crucial to their numerical efficiency. The results of the performed numerical tests with grid spacings down to $25$ m are very promising. However, a direct comparison with the Instructed Glacier Model (IGM) (Jouvet et al., 2021; Jouvet and Cordonnier, 2023) has

not yet been initiated.

Conceptually, the restriction to the SIA is the main limitation of the new scheme at present. This limitation particularly concerns sliding at the bed since the physical basis of the respective friction laws is still uncertain. Concerning the friction laws within the SIA framework, the implementation in MinSIA is quite flexible.

Numerically, steep slopes are the main problem. In contrast to thick ice layers with moderately inclined surfaces, the new

scheme is not much better than an explicit scheme for thin layers on steep slopes. This limitation depends on the topography and on the model parameters and becomes increasingly relevant for finer grids. Practically, some loss of performance was observed for steep alpine topographies at $50$ m grid spacing. However, this loss does not affect the improvement compared to explicit schemes strongly.

There are multiple options to improve the performance at steep slopes even without extending the numerical scheme funda-

mentally. Adding a simple model for avalanches that avoids the accumulation of ice at steep slopes is one option. Alternatively, it would be possible to reduce the sliding velocity at steep slopes, which would enforce thicker ice layers. However, there might be no unique best solution since the reduced performance at steep slopes depends on several conditions. Therefore, addressing this problem is left as a task for future development.

*Code and data availability.*  All codes are available in a Zenodo repository at https://doi.org/10.5281/zenodo.15362846 (Hergarten, 2025b)

and can be redistributed under the GNU General Public License. This repository also contains data obtained from the numerical simulations. Interested users are advised to download the most recent version of the MinSIA software from http://hergarten.at/minsia (Hergarten, 2025a).

*Video supplement.*  A movie which illustrates the Black Foerst test scenario is available at http://hergarten.at/minsia/examples (Hergarten, 2025a).





**Appendix A: The flux density for deformation**

Let us, for simplicity, assume that the ice surface is parallel to the bed with a slope angle $\beta$ and that the $z$-coordinate is
perpendicular to the surface with $z = 0$ at the bed. Then the shear stress acting on a surface-parallel plane is given by the
downslope component of the weight of the ice column above $z$,

$$\tau(z) = \rho g \sin\beta \, (\kappa - z) \tag{A1}$$

for $0 \le z \le \kappa$, where $\rho g$ is the specific weight of the ice and $\kappa$ the thickness perpendicular to the surface. According to Glen's
flow law, the strain rate is

$$\frac{\partial}{\partial z} v_{\mathrm{d}}(z) \propto \tau(z)^n \propto \left(\sin\beta \, (\kappa - z)\right)^n, \tag{A2}$$

where $v_{\mathrm{d}}(z)$ is the respective velocity. This velocity is obtained by integration,

$$v_{\mathrm{d}}(z) = \int_0^z \frac{\partial}{\partial \zeta} v_{\mathrm{d}}(\zeta) d\zeta \propto \left(\sin\beta\right)^n \left(\kappa^{n+1} - (\kappa - z)^{n+1}\right). \tag{A3}$$

Integrating the velocity yields the flux density

$$q_{\mathrm{d}} = \int_0^\kappa v_{\mathrm{d}}(z) dz = f_{\mathrm{d}} \left(\sin\beta\right)^n \kappa^{n+2}, \tag{A4}$$

where $f_{\mathrm{d}}$ was introduced as a factor of proportionality. Expressing $\kappa$ in terms of the vertical thickness $h$ according to $\kappa = h\cos\beta$
yields the relation

$$q_{\mathrm{d}} = f_{\mathrm{d}} h^{n+2} \left(\cos\beta\right)^{n+2} \left(\sin\beta\right)^n = f_{\mathrm{d}} h^{n+2} \left(\cos\beta\right)^{2n+2} \left(\tan\beta\right)^n, \tag{A5}$$

which is the scalar version of Eq. (3) for $n = 3$.

**Appendix B: The flux density for sliding at the bed**

Starting point is the relation

$$v_{\mathrm{s}} \propto \tau^{\frac{n+1}{2}} \frac{\tau}{\sigma - p} \tag{B1}$$

(Cuffey and Paterson, 2010, Eqs. 7.11. and 7.17) for the sliding velocity $v_{\mathrm{s}}$, where $\sigma$ is normal stress and $p$ water pressure.
While the first term describes the original relation proposed by Weertman (1957), the second term is the bed separation index
introduced by Bindschadler (1983). Under the simplification that the fluid pressure $p$ is proportional to the normal stress $\sigma$, the
proportionality of Eq. (B1) persists without $p$ in the denominator. According to Eq. (A1) for $z = 0$, the shear stress at the bed
is

$$\tau = \rho g \sin\beta \kappa. \tag{B2}$$





For the normal stress, the component of gravity perpendicular to the surface must be taken into account, which yields

$$\sigma = \rho g \cos\beta \kappa \tag{B3}$$

Inserting the stresses into Eq. (B1) yields

$$v_s \propto (\kappa \sin\beta)^{\frac{n+1}{2}} \tan\beta \tag{B4}$$

and for the flux density

$$q_s = \kappa v_s = f_s \kappa (\kappa \sin\beta)^{\frac{n+1}{2}} \tan\beta = f_s \kappa^{\frac{n+3}{2}} (\cos\beta)^{\frac{n+1}{2}} (\tan\beta)^{\frac{n+1}{2}} \tan\beta = f_s h^{\frac{n+3}{2}} (\cos\beta)^{n+2} (\tan\beta)^{\frac{n+3}{2}} \tag{B5}$$

with $f_s$ as a factor of proportionality. This equation is the scalar version of Eq. (6) for $n = 3$.

*Author contributions.* S.H. developed the numerical scheme and the codes, performed the tests and wrote the paper.

*Competing interests.* The author declares that there is no conflict of interest.



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
