# Peer review of "MinSIA v1: a lightweight and efficient implementation of the shallow ice approximation"

_EGUsphere, 2025_

## Referee Comment (RC1)

**Reviewer report EGUSPHERE-2025-2242**

**MinSIA v1: a lightweight and efficient implementation of the shallow ice approximation**

Daniel Moreno-Parada

July 31, 2025

The current work explores the numerical benefits of a new algorithm to solve the Shallow Ice Approximation (SIA). It combines a semi-implicit time-stepping with a dynamic smoothing to deal with the nonlinear flow velocity. The scheme is only implemented in MATLAB, as the Python version is still under development. Numerical tests are performed to study the numerical stability in terms of the smoothing factor and timestep, focusing on the potential staircase oscillations and mass conservation.

Overall, I find that the message of the paper is successfully conveyed. Nevertheless, I have a number of major remarks that I would like the author to address and further elaborate on the manuscript. In particular, the work falls short in certain aspects that I have listed in the Major Remarks section. Lastly, I have included a short list with other minor remarks at the end of the document.

**1 Major remarks**

- **Analogy with Airy's linear theory.** The author did not provide any physical justification of the analogy with Airy's wave theory: a linearised description of the propagation of gravity waves on the surface of a homogeneous fluid layer. A priori, this problem fundamentally differs from the SIA description of glaciar ice tackled by MinSIA. In my view, there are several flaws:

  - Airy's exponential decay is a consequence of water wave kinematics (Dingemans, 1997), not directly transferable to ice sheet mechanics. How is the horizontal smoothing in ice surface justified by the exponential decay along the vertical dimension in Airy's linear theory? Why larger ice thicknesses imply a larger length scale of smoothing?
  - Ice behaves as a non-newtonian fluid, not considered in Airy's theory. Viscous dissipation in ice dampens short-wavelength perturbations, altering how surface effects propagate.

  In summary, the analogy lacks physical justification. Ice thickness does not play the same role as wavelength in water waves decay, unless otherwise shown by the author. With the current description, the smoothing appears as an heuristic fix (as stated by the author: "*the thickness-weighted average avoids artifacts at the boundaries of ice-covered area*"), not a physics-derived approach. I encourage the author to either provide a detailed physical justification of the analogy or remove the reference to water wave propagation, thus re-framing the motivation.

- **Inadequacy of SIA.** As described by Greve and Blatter (2004), the SIA is justified for large ice sheets where conditions generally vary little over horizontal distances (i.e., 5-10 times the local ice thickness). However, this is not often the case in mountain glaciers, where longitudinal and transverse coupling of stresses are important and should therefore be considered. To illustrate this inadequacy, Fig. 7.4 (Greve and Blatter, 2004) shows a scatter plot comparison of the First Order Approximation and the SIA velocities for Haut Glacier d'Arolla: the latter underestimates the velocities for values below 10 m/yr, and vice versa for velocities above this threshold.

  Given that "*The purpose of this paper is to challenge the hypothesis that the potential for further improvements in computational efficiency with classical numerical methods is limited (Jouvet et al., 2021)*", I encourage the author to choose a more suitable stress balance approximation and discuss the differences with a more sophisticated description. If the SIA is nonetheless correct, then a simple plot as Fig 7.4 in Greve and Blatter (2004) will suffice to justify and discuss the choice over the domain. Note that the work of Jouvet et al, (2021) employed PISM output to train the emulator, capturing the physics contained in longitudinal stresses present in the SSA. It would be also convenient to discuss how the second-order SIA peforms in this context (e.g., Ahlkrona et al, 2013.)

- **Lack of smooth-free reference.** As stated by the author: "*It would be desirable to use a simulation without smoothing as a reference scenario*". There is an inevitably trade-off between smoothness and numerical stability. I consider that it is thus mandatory to have a reference simulation to quantify the deviation from the smooth-free solution. As stated in the paper: "*the smoothing factor was set to f = 0.25 as a reference*". Why is it so? This number seems arbitrary without further justification. In fact, it corresponds to a strongly oscillating result (Fig. 4) that need a nearly $10^8$ m$^3$ of ice in order to keep the ice surface consistent (Fig. 5). Please, provide detailed physical justification on why $f = 0.25$ should be a reference value.

- **Python implementation.** As stated by the Editor, MATLAB is not the most "open software", so it would be convenient to provide the Python software under development and discuss the performance in the manuscript. In this line, NVIDIA recently announced the cuNumeric library, a drop-in replacement for the NumPy library that allows to run on multi-core CPUs, single or multi-GPU nodes, and even multi-node clusters without changing your Python code. Operations are executed by Legate's task engine and accelerated on one or many NVIDIA GPUs (if no GPU is present, on all CPU cores). Linking with the previous comment, as the author justify the absence of smooth-free reference simulations as a results of prohibiting computational costs, this apparent issue could be potentially overcome by using cuNumeric library. Either way, since the main focus of the paper is to show that there is still room for improvement in computational efficiency with classical numerical methods, I encourage the author to include a section where parallel performance is elaborated.

- **Missing comparison with measured glacier velocities.** I consider that the perturbation induced by smoothing should be rather framed in the context of observed glacier velocities. It is expected that timestep and spatial resolution will impact the presence of numerical artefacts such as the oscillations discussed, but a comparison with observed velocities is fundamental. If this paper aims at keeping "*classical numerics competitive*", it requires some sort of validation with measured velocities. My suggestion would be to compare MinSIA results with observed values as Jouvet et al. (2021). This will serve as a validation test to quantify to what extent smoothing perturbs the velocity field.

- **Overstated stability claim.** Line 107 of the manuscript reads: "*This semi-implicit scheme already ensures stability for arbitrary time increments δt*". This claim is quite

strong and generally incorrect for this specific scheme. If $D$ changes rapidly in space or time, evaluating it explicitly can lead to instabilities for large $\delta t$. The scheme is not unconditionally stable. In fact, the author later states in the paper that special focus is needed on the timestep to avoid numerical oscillations (see my next comment) and even discusses situations where the CFL criterion is relevant. Please, revise other vague statements regarding numerical stability.

- **Focus on "time increment" $\delta t$.** The paper reads that: "*[...] the systematic error in the volumetric balance is negligible compared to the immediate effect of the staircase oscillations on accuracy. So focus should be on limiting $\delta t$ to avoid these oscillations*". Can we simply conclude that, given the numerical nature of a semi-implicit scheme, the stability is determined by the timestep? (and therefore contradicts the pervious claim that the semi-implicit scheme ensures stability for arbitrary timesteps). If so, a fully implicit scheme would overcome this issue? Further experiments are needed to support or reject this hypothesis.

- **Timestepping and redundant sections.** I would suggest merging Section 4.2 (The maximum time increment) and 5 (Finding the best time increment) in a single section. All numerical results regarding timestepping should be described and discussed therein.

  Moreover, I find a great amount of effort on the present work while lacking some important points. For instance, Cheng et al. (2017) introduced an adaptive time step control for simulations of the evolution of ice sheets using Elmer/Ice (Gagliardini et al., 2013). Semi-implicit and fully implicit methods are compared for a number of discretization stencils. I consider that if the problem of "finding the best time increment" is to be tackled, the paper should dive into the predictor-corrector (among others) approaches and showcase the performance in MinSIA.

- **Figures:**

  - Figure 1. This plot is hard to interpret: colour bar is missing, legend is missing, spatial scale is missing, inset with geographical zoom-out is also missing. Please, improve the figure so that they are as much self-explicative as possible.

**2 Minor remarks**

- It would be very convenient to include explicit discretization schemes implemented in MinSIA. Section 3 (Numerical scheme and implementation) elaborates on the smoothing algorithm, but the finite volume and upstream diffusivity schemes are not given in the manuscript. An appendix with the explicit discretization is beneficial for reproducibility and future comparison.

- I wonder how the CFL criterion looks like overlaid in Fig. 7. It is illustrative to show the timestep restriction imposed by the CFL criterion for different resolutions and smoothing values. Moreover, Fig. 7 should also include the deviation from the smooth-free reference simulation. Large values of the smoothing factor could imply unrealistic velocity fields.

- Line 5: What does this mean: "*MinSIA5 is designed for simulations with several million nodes on standard desktop PCs*"? What does the author mean by *node* here? Regular desktop PCs usually have only $\sim$ 16-32 CPU cores.

- Lines 264-270. This paragraph could be synthesized by plotting the computing time as a function of different parameters (e.g., $\delta t$, $\delta x$, $f$, etc.). In log-scale, the slope of the linear fit will show the exponential dependency.

**References**

- Dingemans, Maarten W. (1997) Water wave propagation over uneven bottoms. part 1: Linear Wave propagation. Singapore: World Scientific.

- Cheng, Gong, Per Lötstedt, and Lina von Sydow. "Accurate and stable time stepping in ice sheet modeling." Journal of Computational Physics 329 (2017): 29-47.

- Gagliardini, O., Zwinger, T., Gillet-Chaulet, F., Durand, G., Favier, L., de Fleurian, B., Greve, R., Malinen, M., Martín, C., Råback, P., Ruokolainen, J., Sacchettini, M., Schäfer, M., Seddik, H., and Thies, J.: Capabilities and performance of Elmer/Ice, a new-generation ice sheet model, Geosci. Model Dev., 6, 1299–1318, `https://doi.org/10.5194/gmd-6-1299-2013`, 2013.

- Ahlkrona, J., Kirchner, N., and Lötstedt, P.: Accuracy of the zeroth- and second-order shallow-ice approximation – numerical and theoretical results, Geosci. Model Dev., 6, 2135–2152, https://doi.org/10.5194/gmd-6-2135-2013, 2013.

---

## Referee Comment (RC2)

**Review egusphere 225-2242:**

**MinSIA v1: a lightweight and efficient implementation of the shallow ice approximation**

**General Impression**

The submitted manuscript describes the implementation of an ice-flow model based on the Shallow Ice Approximation (SIA) formulated using the vertically integrated mass balance and implemented in MATLAB. The manuscript further discusses measures in spatial smoothing and time-discretization that had to be taken to stabilize the method in order to produce simulations using synthetic forcings of glaciations over selected regions of the Alps and the Black Forest, the latter used as the benchmark case presented in this study.

Whereas the description of the model seems to be almost complete (I ask for a few more details) and implemented along the guidelines of the target journal (GMD), to which general purpose model descriptions fit, I have to express major concerns on the applicability of the model in view of its approximation and the applied slope correction and the evaluation of its performance and reproducibility by the reader, which I try to elaborate in the sections below.

**Critical issues**

Looking at the abstract text, I already was surprised to read about 25 m spatial resolution (assuming it is the one in horizontal direction) in connection with the applied Shallow Ice Approximation (SIA). This seems to have been confirmed within the text. There is a long series of studies on the validity and accuracy of the SIA by several authors (Jóhannesson, 1992; Gudmundsson, 2003; Hindmarsh, 2004 - to list a few) that later were accompanied by numerical investigations comparing full-stress Stokes (e.g., Le Meur et al. 2004, Seddik et al. 2017 ) and/or higher order approximation models (Pattyn et al., 2008) to SIA. The quintessence of all these studies is that **SIA is inaccurate** with respect to three aspects that apply to the studies presented in this manuscript, namely,

- fast sliding (though this is difficult to estimate speeds from the information given)
- steep slopes and increased accumulation
- terrain resolutions significantly below a few multiples of the local ice thickness.

The last point, I would even say, is a validity criterion, as shallowness (which also applies to longitudinal gradients to be resolved) is the fundamental parameter behind the expansions of the Stokes equation leading to the zeroth-order equations, the SIA (Hindmarsh, 2004 ). In other words, based on the above mentioned literature, to me a resolution below about a few times the local ice-thickness is in violation with the principle assumption that go into the derivation of the approximation applied in this study.  I am not surprised by the instabilities arising from resolving steep glaciers at 25 m with SIA. I would also like to see a mathematical argument to whether the ad-hoc introduced slope correction in equation (3) is consistent with the assumption of the SIA (I think it is not). The author leaves the reader with no means of quantifying the inaccuracies introduced by the choice of the method in combination with the resolution and steepness of the terrain, since the reference run is provided with the method itself. It either would take comparison with observations (not possible for synthetic ELA forcing) or a higher order (if not Stokes) solution to the problem. To me, these issues seem not to be fixable by improved numerical methods, as they are inherent to the approximation applied. Thus, I would see the necessity to demonstrate the mathematical consistency of the presented model with the zeroth order approximation (SIA). Further, I would expect a clear proof of the advantage of running SIA on – how I conclude - 2 orders of magnitude above the with theory consistent mesh resolutions and quantify the errors introduced by steep

slopes/high accumulation/fast sliding to comparison of either observed results or results obtained with to the task suitable approximations to or the Stokes equations. This not necessarily has to be done on domains as big as whole parts of the Alps.

The second concern is linked to the main motivation given by the author to provide a computationally fast method to the community in order to compete with machine learning (ML) algorithms. My deviating opinion on this competition aside (see in specific comments), I see the following **issues** concerning the **description of model performance and reproducibility**, namely

- the proprietary package needed to run the implementation (MATLAB) prohibiting reproduction of the results for readers without a valid license
- the lack of detailed description on applied parallel computation paradigms (if any), like OpenMP (threading and SIMD) or even distributed memory, like MPI. Can the setup perhaps even utilize accelerators?
- the detailed justification on the choice of the methods (one direct and one Krylov) used to solve the sparse matrix problem

I will in detail refer to these items in the specific comments section.

**Specific Comments**

I indicate the line number as given in the egusphere-document. I display quotes from the manuscript in a greenish colour. It will be pointed out if a comment relates to a major point of critics (previous section)

- Line 2: This paper aims at keeping classical numerics competitive in this field. This is merely a comment conveying my divergent opinion: I do not think that the competition (if there is, as I rather see them as complementary) between machine learning (ML) algorithms and process based models (*classical* in terms used here) is about speed. In my opinion, it is about accuracy and physical completeness. ML reproduces what it has been trained with, and it does this very fast and efficient. The particular reason why these techniques (or at least in my opinion should be) utilized is exactly to speed things up. On the other hand, ML is restricted to a given setup (fixed set of geometry and/or processes) defined by the training set. These surrogate models need new training (which tends to be computationally intensive, so no free lunch also here) to introduce new physics, often even if moving to another topography. That – in my opinion – is where the strength of process models lies – they easier can introduce improved physics and they are more generally applicable.  I, though, think it is futile to try to compete in terms of speed with a ML implementation using the same setup in the training set as in a process-based model (of which complexity it ever may be), in particular, if the training phase of the ML is left out of the comparison.
- Line 5: … a lightweight implementation of the new scheme in MATLAB. This relates to the second major point before. In my opinion, it would have been better to wait for the Python implementation to be ready before publishing. At this point in time, the reader needs a licensed software (*MATLAB*) to reproduce results and is left without any insight on the performance of the Python implementation to come. To palliate the first problem, perhaps the author can test with the open source software *Octave* and report to what extend this can be used in replacement to *MATLAB* to run your current version of the code?
- Line 16: Three-dimensional simulations of the Stokes equations with a free surface (as implemented, e.g., in the model Elmer/Ice) and the shallow ice approximation (SIA) are end-members in the hierarchy of ice-flow models.  May I in this connection point out that there are existing Stokes simulations of the Western part of the Alps (in fact, including Black Forest, which

was not part of the investigation but the domain) spanning several thousand years around the LGM (Cohen et al, 2018 and 2023).

- Line 17: The SIA considers vertically averaged velocities, assumes hydrostatic pressure, and neglects all stresses arising from horizontal shearing. To my knowledge, SIA defines horizontal velocities as a function of the vertical coordinate (Greve and Blatter, 2009), but only its force-balance considers vertically integrated stresses (expressed in terms of integrals of the vertically varying horizontal velocity field along columns). So, the fluxes, but not the velocities are vertically integrated variables. Could it be that in the presented application the fact that one would have to perform a quadrature of the force balance in each column is circumvented by reducing the solution to the mass-balance and the introduction of the pre-factors $f_d$ and $f_s$ therein? If so, I would ask to explain what assumptions the replacement of the vertical integrals by – what occurs to me – fixed pre-factors are implicitly introduced by this procedure (e.g. on vertical distributions of enhancement factors, temperatures).

- Line 18: In between, there are two-dimensional approaches that account for these stresses, … . Even Stokes can be solved in two dimensions (flow line). I would suggest to write depth-integrated instead.

- Line 26: The still limited performance of two-dimensional models arises from a combination of assuming hydrostatic pressure and the explicit time-stepping scheme implemented in almost all contemporary models (Bueler, 2023). Like before, I would suggest using depth-averaged or integrated models, instead. Further I would ask to explain what performance is referred to? Pure computational performance or beyond that also in terms of accuracy and stability? In view of the latter (which links to my primary point of criticism), I would drop the "still", as to me it implies that the instabilities/inaccuracies described in this manuscript are something that could be overcome. In my opinion, they are built-in, as models based on hydrostatic approximation are not capable to resolve horizontal gradients in stresses that act on scales smaller than a few multiples of the ice thickness and assume small slopes (Hindmarsh, 2004) .

- Line 62, eq. (3) and line 68 and eq. (5): The factor involving the cosine of the slope angle β of the ice surface with … is a simple correction for steep slopes. I will come back to this later also in the appendix, but to me it appears that introducing this slope correction violates a principal assumption of the SIA that the surface normal in lowest order points exactly into the vertical direction which by shallowness also would demand that $\cos\beta \approx 1 \rightarrow \beta \approx 0$ . In the other extreme, for $\beta \rightarrow \pi/2$ the 5th and 8th power of $\cos\beta$ in (9) will result in a vastly vanishing pseudo diffusivity and (8) basically reduces to the geodetic mass balance. In other words, mass transport shuts down on steep slopes. That in my view is artificial and covers the fact that the hydrostatic pressure approximation is not valid on slopes significantly deviating from zero (Hindmarsh, 2004) - see point one in major critics.

- Line 67: The factor $f_d$ mainly depends on temperature in reality. This factor should also contain some contributions from the vertical integral of the force balance, including the temperature dependence and enhancement factors. As demanded before, I think for completeness, the exact composition of $f_d$ as well as $f_s$ leading to the constant values reported in the text should be written out and explained – if not in the main part, at least in the Appendix. Further, as this seems to be the only mentioning of temperature, which, by the Arrhenius factor (Greve and Blatter, 2009) has a strong influence on the ice viscosity and hence the flow: I would kindly ask how the model – if at all - treats temperature variations in the ice, in particular as I would think that during a glaciation of the Alps the thermal ice conditions significantly vary, both, spatially and in time? Would the presented model be in principle capable of including advective temperature transport?

- Line 70: The correction is explained in the derivation of Eq. (3) in Appendix A. It should, however, be mentioned that it is less elaborate than the correction for rapid mass movements developed by Savage and Hutter (1989) and only exact if the ice surface is parallel to the bed. In particular, it does not capture the effect of steep walls in a valley. I discuss this in detail later where the method is explained in the Appendix. As mentioned before, I think the cosine in (3) is in violation of the basic assumptions behind SIA. I also do not see the parallel slab problem of a creeping shear thinning fluid in a direct connection to the Savage-Hutter theory, which is an expansion of the depth-averaged Navier-Stokes equations (i.e., including acceleration terms) for yielding granular flows, neglecting terms of order $O(\epsilon^{3/2})$ of the aspect ratio $\epsilon$, and in contrary to the SIA, usually already defined in a locally rotated coordinate system (e.g., Zwinger et al. , 2003).
- Line 80: It should, however, be emphasized that modeling sliding along the bed is still one of the major challenges in this field. It is even questionable whether this process can be described well by the SIA or whether lateral stress components have to be taken into account. Please, refer to literature where these topics are addressed. For the latter: It is established (see literature cited in the major points section) that SIA is unsuited for fast sliding conditions.
- Line 89: Formally, it looks like a nonlinear diffusion equation, although it is mathematically not a diffusion equation due to the occurrence of $\nabla s$ in $D$. Can you please elaborate: If not a diffusion equation, what is it then?
- Line 120: The upstream scheme ensures D = 0 in this situation and thus avoids a permanent systematic increase in ice volume. What about the considerations on the second term of the r.h.s. of (8) at the contact line in order to avoid negative heights?  There is often negative net mass balance, $r$, at the front of the glacier. Is it masked out? Generally speaking, to stay consistent in mass balance one should solve the mass balance or in higher order models the kinematic boundary condition together with the inequality for positive or zero flow-heights (see, e.g., Gagliardini et al. 2013, section 6.5) , which in my opinion should be applicable to the scheme applied in this manuscript.
- Line 128: Therefore, a dynamic smoothing of the slope of the ice surface in the diffusivity $D(h,|\nabla s|)$ is the central novel idea in MinSIA. The idea was originally motivated by Airy's linear theory of ocean waves (gravity waves in an inviscid fluid with a free surface). According to this theory, oscillations in pressure and particle velocity decrease exponentially with depth below the surface. The respective depth of penetration is proportional to the wavelength. Based on this result, MinSIA uses a smoothed version of $|\nabla s|$ with the length scale of smoothing being proportional to the ice thickness h. I am not able to see a connection from a theory based on gravity waves in Newtonian fluids to the mechanical behaviour of a (strongly non-linear and highly viscous) shear thinning free-surface thin-film Stokes problem. My theory:  this smoothing algorithm works because it re-introduces the basic principle of thin-film approximations to resolve horizontal gradients with respect to a large aspect ratio (i.e., over several ice-thicknesses), even if run on mesh resolutions that apparently violate this assumption. To probe that theory, I suggest to run the simulation on a to the SIA appropriate mesh resolution (>1000 m range) without smoothing and report on instabilities of such a run. I would also ask to include a clear motivation to opt for such small horizontal resolutions of 25 m – 100 m in connection with SIA. Thinking of the main motivation presented in the text, a fast solution, I would even say that increasing the resolution should work exactly in this direction as the problem size gets smaller. This links to the first in the main point of critics.
- Line 163: The preconditioned conjugate gradients (PCG) method is used for solving the linear equation system. An incomplete Cholesky factorization is used as a preconditioner, whereby the version that compensates dropped nondiagonal elements at the respective diagonal elements

turned out to be particularly suitable. A direct solver is also implemented for testing. As mentioned in the second point of major critics. I would ask for a more detailed explanation of this particular choice. What method was used for the direct solver? Were these methods utilizing specific features of the hardware (shared-memory or distributed memory parallelism, vector units, accelerators, if present)? What can the reader expect if this model is run on other, more modern platforms than the CPU that was reported in the text? Are there special license conditions to utilize the mentioned methods in MATLAB?

- Line 187: As a first test, the accuracy of the solutions for different values of the smoothing factor $f$ with respect to the reference solution ($f$ = 0.25, $\delta t$ = 64 yr) is measured. In relation to the major point of critics, I do not think that comparison to a result obtained with the method is able to show the accuracy of the method itself. To me, it merely shows the sensitivity of the method to a change in numerical parameters and resolutions.

- Line 209: Since even explicit schemes are stable for sufficiently small time increments, the occurrence of the staircase oscillation must be related to both the smoothing factor $f$ and the time increment $\delta t$. I would like to see a proof on the statement of stability for small timesteps. Either by citing literature or running the model (at least for some time) with no smoothing on the 25 m resolution. What I think, is that the oscillations are a reaction of the SIA to significant undulations with a length-scale way below the ice-thickness. Consequently, I would conclude that the oscillations are natural to the SIA (if run on resolutions that violate the shallowness assumption) and damped out by the smoothing, similar to as they would by running the SIA without smoothing on coarser meshes (which I already asked you to test) that resolve horizontal scales above the typical ice-thicknesses.

- Line 267: The simulations over the entire time span with $\delta t$ = 0.25 yr took 35400 s for $\delta x$ = 25 m, 6740 s for $\delta x$ = 50 m, and 1290 s for $\delta x$ = 100 m on an Intel Core i5-7600 CPU (3.50 GHz) from 2018. This, to me, seems to be the only information on how some sort of performance evaluation has been done. In my view, this is not enough such that the reader can get a clear picture what to expect if running the model themselves. I would ask to specify the utilization of the CPU (single core or multiple cores, as there seem to be 4 included in the reported model). If multiple cores have been utilized, please report on what parallel paradigm was deployed (shared or distributed memory). How much memory was available in the test platform and how was the saturation of the available memory at the runtime? This links to the second point of my major critics.

- Line 288: Figure 8 illustrates the weakness of the scheme at steep slopes for $\delta x$ = 25 m and $\delta t$ = 0.25 yr. I would rather say it reveals the inapplicability of the approximation than a weakness in the numerical scheme (see major point of critics).

- Line 305: At present, a MATLAB implementation of MinSIA is available under the GNU General Public License. A Python implementation is under development. As mentioned under the major points and before in this section – despite the setup being shared with an open license – the proprietary nature of the software needed to run it imposes a big hurdle for people to reproduce these findings. This all would have been no problem if the publication would have been postponed to the point in time when the announced Python version would have been available.

- Line 334: A 2-element array corresponds to $r_{max}$ = ∞ and a scalar value additionally to $g-$= $g+$. I am not able to understand the meaning behind this sentence. Can you please elaborate? Do you allow for infinitely large accumulation rates?

- Line 390: Let us, for simplicity, assume that the ice surface is parallel to the bed with a slope angle β and that the z-coordinate is perpendicular to the surface with z = 0 at the bed. I think this is the culprit of the discrepancy about the bed-slope correction I raised in the main issues. The parallel slab example (see Greve and Blatter, 2009), which seems to be adapted here, in my view, cannot

be intermixed with the derivation of the SIA. The basic assumption behind the SIA is a directly into the vertical direction pointing surface normal (Greve and Blatter, 2009). The resulting full alignment with the negative direction of the gravity is a necessary condition for the zero Cauchy stress vector at the surface leading to identically vanishing stress-components and pressure, which constitutes another necessary condition for the vertical integration of the shear stresses (Greve and Blatter, 2009) in SIA.  If the integration in the vertical direction is done from bedrock, $b$, to the free surface $h$, one obtains the shear stress at the bedrock as $\tau = \rho g\, h\, \nabla s$, where $h = s - b$, i.e., the total ice-depth in vertical direction. Thus, in SIA the shear stress is imposed by the hydrostatic pressure gradient induced by the horizontal change in surface elevation (with no influence of the bedrock slope at all), whereas in the parallel slab, i.e., equation (A1), the shear stress is the result of the downslope weight component of a in the rotated reference frame bed-parallel surface with in this rotated reference frame identically vanishing gradients of the free surface. As I see it, the author is setting (A1) identical to the bed-shear stress as defined in SIA, which I claim violates the basic assumptions of the SIA.  If the author disagrees, I would ask to include a detailed derivation of the SIA (starting from the Stokes equations) in a rotated reference frame and show that the derived equations still comply with the lowest order approximation, even for large bedrock angles, as they occur in the studied topographies of the Alps and even the Black Forest.

**Typos and technical corrections**

Line 245: PGC -> PCG

Line 281: … which  also is still stronger than …

Line 396: $\tau(z)^n$ suggestion to change to $\tau^n(z)$

Line 409: suggestion: first term -> first multiplier on the right-hand-side

**References**

Cohen, D., F. Gillet-Chaulet, W. Haeberli, H. Machguth, and U.H. Fischer, 2018. *Numerical reconstructions of the flow and basal conditions of the Rhine glacier, European Central Alps, at the Last Glacial Maximum*, The Cryosphere, **12**, 2515-2544, doi: 10.5194/tc-12-2515-2018

Cohen D., G. Jouvet, T. Zwinger, A. Landgraf and U. H. Fischer, 2023. *Subglacial hydrology from high-resolution ice-flow simulations of the Rhine Glacier during the Last Glacial Maximum: a proxy for glacial erosion,*. E&G Quaternary Sci. J., **72**, 189–201, doi:10.5194/egqsj-72-189-2023

Gudmundsson, G. H.,  2003, *Transmission of basal variability to a glacier surface*, J. Geophys. Res., 108(B5), 2253, doi:10.1029/2002JB002107.

Greve and Blatter, 2009. Dynamics of Ice Sheets and Glaciers, Springer

Hindmarsh, R.C.A. , 2004, *A numerical comparison of approximations to the Stokes equations used in ice sheet and glacier modelling*, J. Geophys. Res.: Earth Surface, 109, F01012, doi:10.1029/2003JF000065

Johannesson, T., 1992, *The landscape of temperate ice caps*, Ph.D. thesis, University of Washington, Seattle

Le Meur E., O. Gagliardini, T. Zwinger, J. Ruokolainen, 2004, *Glacier flow modelling: a comparison of the Shallow Ice Approximation and the full-Stokes equation*, C. R. Physique **5**, 709-722, doi:10.1016/j.crhy.2004.10.001

Pattyn, F., Perichon, L., Aschwanden, A., Breuer, B., de Smedt, B., Gagliardini, O., Gudmundsson, G. H., Hindmarsh, R. C. A., Hubbard, A., Johnson, J. V., Kleiner, T., Konovalov, Y., Martin, C., Payne, A. J., Pollard, D., Price, S., Rückamp, M., Saito, F., Souček, O., Sugiyama, S., and Zwinger, T., 2008, *Benchmark experiments for higher-order and full-Stokes ice sheet models (ISMIP–HOM)*, The Cryosphere, 2, 95–108, doi:10.5194/tc-2-95-2008

Seddik, H., R. Greve, T. Zwinger, and S. Sugiyama, 2017. *Regional modeling of the Shirase drainage basin, East Antarctica: full Stokes vs. shallow ice dynamics*, The Cryosphere, **11**, 2213-2229, doi:10.5194/tc-11-2213-2017.

Zwinger, T., Kluwick, A., and Sampl, P., 2003, *Numerical Simulation of Dry-Snow Avalanche Flow over Natural Terrain*. In: Hutter, K., Kirchner, N. (eds) *Dynamic Response of Granular and Porous Materials under Large and Catastrophic Deformations*. Lecture Notes in Applied and Computational Mechanics, vol 11. Springer, Berlin, Heidelberg, doi:10.1007/978-3-540-36565-5_5

---

## Author Comment (AC1)

**Analogy with Airy's linear theory.** *The author did not provide any physical justification of the analogy with Airy's wave theory: a linearised description of the propagation of gravity waves on the surface of a homogeneous fluid layer. A priori, this problem fundamentally differs from the SIA description of glacier ice tackled by MinSIA. In my view, there are several flaws:*

- *Airy's exponential decay is a consequence of water wave kinematics (Dingemans, 1997), not directly transferable to ice sheet mechanics. How is the horizontal smoothing in ice surface justified by the exponential decay along the vertical dimension in Airy's linear theory? Why larger ice thicknesses imply a larger length scale of smoothing?*

- *Ice behaves as a non-newtonian fluid, not considered in Airy's theory. Viscous dissipation in ice dampens short-wavelength perturbations, altering how surface effects propagate.*

*In summary, the analogy lacks physical justification. Ice thickness does not play the same role as wavelength in water waves decay, unless otherwise shown by the author. With the current description, the smoothing appears as an heuristic fix (as stated by the author: "the thickness-weighted average avoids artifacts at the boundaries of ice-covered area"), not a physics-derived approach. I encourage the author to either provide a detailed physical justification of the analogy or remove the reference to water wave propagation, thus re-framing the motivation.*

I did not claim that the smoothing approach can be formally derived from Airy's theory. It was only used as an example that short-wavelength oscillations do not accelerate the entire column of material as assumed in the shallow water/ice approximation, but typically have a limited depth of penetration. The depth of penetration is proportional to the wavelength only for linear problems (Airy's theory, but also for elastic materials). Qualitatively, however, it should also remain valid for nonlinear flow laws.

**Inadequacy of SIA.** *As described by Greve and Blatter (2004), the SIA is justified for large ice sheets where conditions generally vary little over horizontal distances (i.e., 5-10 times the local ice thickness). However, this is not often the case in mountain glaciers, where longitudinal and transverse coupling of stresses are important and should therefore be considered. To illustrate this inadequacy, Fig. 7.4 (Greve and Blatter, 2004) shows a scatter plot comparison of the First Order Approximation and the SIA velocities for Haut Glacier d'Arolla: the latter underestimates the velocities for values below 10 m/yr, and vice versa for velocities above this threshold.*

*Given that "The purpose of this paper is to challenge the hypothesis that the potential for further improvements in computational efficiency with classical numerical methods is limited (Jouvet et al., 2021)", I encourage the author to choose a more suitable stress balance approximation and discuss the differences with a more sophisticated description. If the SIA is nonetheless correct, then a simple plot as Fig 7.4 in Greve and Blatter (2004) will suffice to justify and discuss the choice over the domain. Note that the work of Jouvet et al, (2021) employed PISM output to train the emulator, capturing the physics contained in longitudinal stresses present in the SSA. It would be also convenient to discuss how the second-order SIA performs in this context (e.g., Ahlkrona et al 2013.)*

It is not my intention to convince anyone to use the shallow ice approximation. The approach can probably be extended by lateral stresses. The simplest version would treat the respective derivatives of the velocity as an explicit component in the semi-implicit scheme. However, I just propose an efficient numerical scheme for the simplest glacial flow law.

**Lack of smooth-free reference.** *As stated by the author: "It would be desirable to use a simulation without smoothing as a reference scenario". There is an inevitably trade-off between smoothness and numerical stability. I consider that it is thus mandatory to have a reference simulation to quantify the deviation from the smooth-free solution. As stated in the paper: "the smoothing factor was set to $f = 0.25$ as a reference". Why is it so? This number seems arbitrary without further justification. In fact, it corresponds to a strongly oscillating result (Fig. 4) that need a nearly 108 m3 of ice in order to keep the ice surface consistent (Fig. 5). Please, provide detailed physical justification on why $f = 0.25$ should be a reference value.*

To be honest, I do not understand how you obtained the "strongly oscillating result" from Fig. 4 and "need a nearly $10^8$ m$^3$ of ice in order to keep the ice surface consistent" from Fig. 5. For $f = 0.25$ and $dt = 1/64$ yr (the reference), the 95 % oscillation (exceeded in 5 % of all time steps) is about $2 \times 10^{-5}$ m and the absolute maximum oscillation about $5 \times 10^{-4}$ m. Given all the uncertainties in glacier modeling, I would not consider oscillations of less than 1 millimeter strong. The added volume is always less than $3.5 \times 10^{-5}$ m$^3$ in each time step and less than 5 m$^3$ added over the entire 3000 yr simulation, which seems to be small in relation to the total ice volume of about $5 \times 10^{11}$ m$^3$. I tried to explain why simulations with smaller smoothing parameters are too expensive in lines 180-82.

Anyway, I just performed some additional tests. Without smoothing, I need $\delta t = 1/4096$ yr. Starting from $t = 2900$ yr (from the beginning, it would really be too expensive), the simulation takes 3 hours per year. The root mean square deviation in thickness between $f = 0.25$ and no smoothing is about 1.5 cm and the maximum deviation (over all points) about 35 cm. So these deviations are also completely irrelevant compared to all other uncertainties.

**Python implementation.** *As stated by the Editor, MATLAB is not the most "open software", so it would be convenient to provide the Python software under development and discuss the performance in the manuscript. In this line, NVIDIA recently announced the cuNumeric library, a drop-in replacement for the NumPy library that allows to run on multi-core CPUs, single or multi-GPU nodes, and even multi-node clusters without changing your Python code. Operations are executed by Legate's task engine and accelerated on one or many NVIDIA GPUs (if no GPU is present, on all CPU cores). Linking with the previous comment, as the author justify the absence of smooth-free reference simulations as a results of prohibiting computational costs, this apparent issue could be potentially overcome by using cuNumeric library. Either way, since the main focus of the paper is to show that there is still room for improvement in computational efficiency with classical numerical methods, I encourage the author to include a section where parallel performance is elaborated.*

Looks like a good suggestion, although I do not know how deep the cuNumeric library goes into solving linear equation systems.

Concerning the smooth-free reference solution, however, I have only one (even quite old) NVIDIA card available. And if I want to use a smooth-free reference scenario with $\delta t = 1/4096$ yr, I would need simulations with different smoothing factors in order to separate the effect of smoothing from the effect of $\delta t$. Then it would become expensive again and it would still be much effort for deviations of a few centimeters.

Anyway, SciPy seems to be not as good as I expected concerning linear solvers. So I am still struggling with the Python implementation and far off from reaching the same efficiency as the MATLAB version.

**Missing comparison with measured glacier velocities.** *I consider that the perturbation induced by smoothing should be rather framed in the context of observed glacier velocities. It is expected that timestep and spatial resolution will impact the presence of numerical artefacts such as the oscillations discussed, but a comparison with observed velocities is fundamental. If this paper aims at keeping "classical numerics competitive", it requires some sort of validation with measured velocities. My suggestion would be to compare Min-SIA results with observed values as Jouvet et al. (2021). This will serve as a validation test to quantify to what extent smoothing perturbs the velocity field.*

Again, it is not my aim to justify the shallow ice approximation by tuning the parameters to reproduce any measured velocities. Without smoothing, there are considerable oscillations at the surface due to the staircase-like ice surface, and these oscillations vanish with increasing smoothing. Nothing beyond.

**Overstated stability claim.** *Line 107 of the manuscript reads: "This semi-implicit scheme already ensures stability for arbitrary time increments $\delta t$". This claim is quite strong and generally incorrect for this specific scheme. If $D$ changes rapidly in space or time, evaluating it explicitly can lead to instabilities for large $\delta t$. The scheme is not unconditionally stable. In fact, the author later states in the paper that special focus is needed on the timestep to avoid numerical oscillations (see my next comment) and even discusses situations where the CFL criterion is relevant. Please, revise other vague statements regarding numerical stability.*

For this discussion, it would be helpful to know which definition of instability you are referring to. I refer to the definition that the spatial variation in the solution increases with decreasing $\delta x$ at fixed $\delta t$. For this scheme, the oscillation is almost independent of $\delta x$ at fixed $\delta t$. This means that Fig. 4 would look basically the same at larger $\delta x$. The oscillation does also not increase strongly through time – definitely not exponentially as it is typically the case for unstable schemes. The increase in oscillation is even weaker than the increase in maximum thickness. So I would say it is formally unconditionally stable, but with a poor accuracy if smoothing is too weak in relation to $\delta x$ and $\delta t$. So please tell me which definition of instability you are referring to since I cannot "revise other vague statements regarding numerical stability" otherwise.

**Focus on "time increment" $\delta t$.** *The paper reads that: "[dots] the systematic error in the volumetric balance is negligible compared to the immediate effect of the staircase oscillations on accuracy. So focus should be on limiting $\delta t$ to avoid these oscillations". Can we simply conclude that, given the numerical nature of a semi-implicit scheme, the stability is determined by the timestep? (and therefore contradicts the previous claim that the semi-implicit scheme ensures stability for arbitrary timesteps). If so, a fully implicit scheme would overcome this issue? Further experiments are needed to support or reject this hypothesis.*

Also not sure whether I got your point correctly. Yes, there is a sharp transition between "ok" and oscillations leading to a poor accuracy (instability according to your definition), and this transition can be quantified in terms of a maximum $\delta t$ depending on $\delta x$ etc. A fully implicit scheme that overcomes this limitation should presumably be a "genuine" implicit scheme and not an iterative scheme built around the mixed scheme. Bueler (2016, doi 10.1017/jog.2015.3) proposed such a scheme, admitting later (Bueler, 2023, doi 10.1017/jog.2022.113) that "The scaling of the Bueler (2016) implicit SIA solver is not enough better than such an explicit scheme, however." This finding is consistent with my findings on the semi-implicit scheme. At $f = 1$ and $\delta t = 0.25$ yr, the highest Fourier number is $> 1500$. As stated above, we have to decrease $\delta t$ to $1/4096$ yr without smoothing, which reduces the Fourier number to about 1.5. This is still higher than for an explicit scheme, but practically not enough to justify solving a big linear equation system in each step. The same holds for the predictor-corrector schemes you are promoting in the next point. So please let me know what kind of further experiments you would like to see.

**Timestepping and redundant sections.** *I would suggest merging Section 4.2 (The maximum time increment) and 5 (Finding the best time increment) in a single section. All numerical results regarding timestepping should be described and discussed therein.*
*Moreover, I find a great amount of effort on the present work while lacking some important points. For instance, Cheng et al. (2017) introduced an adaptive time step control for simulations of the evolution of ice sheets using Elmer/Ice (Gagliardini et al., 2013). Semi-implicit and fully implicit methods are compared for a number of discretization stencils. I consider that if the problem of "finding the best time increment" is to be tackled, the paper should dive into the predictor-corrector (among others) approaches and showcase the performance in MinSIA.*

The first draft of the manuscript contained such a merged section, but then I decided to separate the practical aspects of finding a "good" $\delta t$ from the numerical analysis (and included some results from the Alpine examples). I still think that it is better to keep this more practical section for those who do not want to go though the entire numerical analysis.
And of course, it would be possible to built an adaptive time step control around the step() method, based on the maximum oscillation. However, I want to keep the implementation is small as possible at the moment.
Furthermore, I have no idea what to write about the predictor-corrector schemes in the reference Cheng et al. (2017). As far as I can see, these schemes do not increase the maximum achievable time increment, but only yield a higher accuracy than an explicit scheme for small $\delta t$.

Figure 1. This plot is hard to interpret: colour bar is missing, legend is missing, spatial scale is missing, inset with geographical zoom-out is also missing. Please, improve the figure so that they are as much self-explicative as possible.

I can try.

It would be very convenient to include explicit discretization schemes implemented in MinSIA. Section 3 (Numerical scheme and implementation) elaborates on the smoothing algorithm, but the finite volume and upstream diffusivity schemes are not given in the manuscript. An appendix with the explicit discretization is beneficial for reproducibility and future comparison.

I can do this, but I feel that it would be a bit trivial. Finite-volume diffusion on a regular grid should be clear for those who get so deep into the implementation. And then taking the diffusivity from the node with the higher surface elevation is also simpler in words than in mathematical symbols.

I wonder how the CFL criterion looks like overlaid in Fig. 7. It is illustrative to show the timestep restriction imposed by the CFL criterion for different resolutions and smoothing values. Moreover, Fig. 7 should also include the deviation from the smooth-free reference simulation. Large values of the smoothing factor could imply unrealistic velocity fields.

Since the velocities are almost independent of the smoothing factor (as long as the oscillations are not too strong), the CFL criterion would be a horizontal line. Roughly $\delta t = 1/20$ year for $\delta x = 25$ m, $\delta t = 1/10$ year for $\delta x = 50$ m, etc. Not sure whether this information provides new insights. And what do you mean with "deviation from the smooth-free reference simulation"? I could include a data point left down for with $f = 0$ (unfortunately, a bit difficult with the logarithmic scale), $\delta t = 1/4096$ yr and the respective values for the other $\delta x$, but I am not sure whether it improves the figure. And again, large values of the smoothing factor have a limited effect on the velocity field and do not imply unrealistic velocity fields.

Line 5: What does this mean: "MinSIA is designed for simulations with several million nodes on standard desktop PCs"? What does the author mean by node here? Regular desktop PCs usually have only $\sim$ 16-32 CPU cores.

Nodes of the finite-volume grid.

Lines 264-270. This paragraph could be synthesized by plotting the computing time as a function of different parameters (e.g., $\delta t$, $\delta x$, $f$, etc.). In log-scale, the slope of the linear fit will show the exponential dependency.

I thought about this, but I have a very limited amount of timing data available. Since the computations were performed on different PCs with different numbers of processes running, I cannot reconstruct the computing times from the metadata of the output files.

---

## Author Comment (AC2)

*Line 2: "This paper aims at keeping classical numerics competitive in this field." This is merely a comment conveying my divergent opinion: I do not think that the competition (if there is, as I rather see them as complementary) between machine learning (ML) algorithms and process based models (classical in terms used here) is about speed. In my opinion, it is about accuracy and physical completeness. ML reproduces what it has been trained with, and it does this very fast and efficient. The particular reason why these techniques (or at least in my opinion should be) utilized is exactly to speed things up. On the other hand, ML is restricted to a given setup (fixed set of geometry and/or processes) defined by the training set. These surrogate models need new training (which tends to be computationally intensive, so no free lunch also here) to introduce new physics, often even if moving to another topography. That – in my opinion – is where the strength of process models lies – they easier can introduce improved physics and they are more generally applicable. I, though, think it is futile to try to compete in terms of speed with a ML implementation using the same setup in the training set as in a process-based model (of which complexity it ever may be), in particular, if the training phase of the ML is left out of the comparison.*

*Line 5: "... a lightweight implementation of the new scheme in MATLAB." This relates to the second major point before. In my opinion, it would have been better to wait for the Python implementation to be ready before publishing. At this point in time, the reader needs a licensed software (MATLAB) to reproduce results and is left without any insight on the performance of the Python implementation to come. To palliate the first problem, perhaps the author can test with the open source software Octave and report to what extend this can be used in replacement to MATLAB to run your current version of the code?*

I share your point of view, but my knowledge about AI-based approaches is not sufficient. In particular, I cannot assess the reliability of the "physics-informed deep learning" version published in 2023. But as a spectator, I see that the AI-based approach has motivated several researchers to consider numerical modeling of ice flow. To me, it seems that the huge numerical effort (I accept that you may have a different opinion what huge means here) of classical numerical models kept researchers off from this field for several years.

In response to the editor's comment on MATLAB, I promised to provide the Python implementation before the final paper is published. In short:

- Everything except for the linear solver: smooth and fast

- Linear solver based on SciPy functions: poor performance

- SciPy CG solver with re-engineered MATLAB preconditioner (incomplete Cholesky with specific treatment of dropped entries: poor performance, even with Numba just-in-time compiling

- Linear solver written in C (same as in MATLAB, but simplified for this specific equation): quite good performance

In the preliminary tests, the Python version with the compiled C solver was even faster than the MATLAB version.

*Line 16: "Three-dimensional simulations of the Stokes equations with a free surface (as implemented, e.g., in the model Elmer/Ice) and the shallow ice approximation (SIA) are end-members in the hierarchy of ice-flow models." May I in this connection point out that there are existing Stokes simulations of the Western part of the Alps (in fact, including Black Forest, which was not part of the investigation but the domain) spanning several thousand years around the LGM (Cohen et al, 2018 and 2023).*

I remember talking with Denis Cohen about his simulations of the Rhine glacier long ago. I also remember that he told that Elmer/Ice full Stokes simulations are not much more expensive than PISM simulations. However, I think that they were still costly in relation to the spatial resolution. But now I am a bit lost what to do with your comment.

*Line 17: "The SIA considers vertically averaged velocities, assumes hydrostatic pressure, and neglects all stresses arising from horizontal shearing." To my knowledge, SIA defines horizontal velocities as a function of the vertical coordinate (Greve and Blatter, 2009), but only its force- balance considers vertically integrated stresses (expressed in terms of integrals of the vertically varying horizontal velocity field along columns). So, the fluxes, but not the velocities are vertically integrated variables. Could it be that in the presented application the fact that one would have to perform a quadrature of the force balance in each column is circumvented by reducing the solution to the mass-balance and the introduction of the pre-factors $f_d$ and $f_s$ therein? If so, I would ask to explain what assumptions the replacement of the vertical integrals by – what occurs to me – fixed pre-factors are implicitly introduced by this procedure (e.g. on vertical distributions of enhancement factors, temperatures).*

The "vertically averaged velocities" are remnant of shallow-water or Savage–Hutter equations, which I am more familiar with. Of course, the velocity is not an independent variable in the SIA, but defined by thickness and gradient of the ice surface. The vertical profile of the horizontal velocity is the integral given in Eq. A3. The fluxes are then obtained by integrating once more (Eq. A4) and the depth-averaged velocity is just flux divided by thickness. As soon as the mechanical properties of the ice become depth-dependent, the proportionalities of Eqs. A2 and A3 get depth-dependent factors. Getting $f_d$ from the vertical profile of the deformation parameter would, of course, become more complicated then. However, I am not sure what "replacement of the vertical integrals" means. The integrals are just solved for constant (not depth-dependent) parameter values.

*Line 18: "In between, there are two-dimensional approaches that account for these stresses, ... ." Even Stokes can be solved in two dimensions (flow line). I would suggest to write depth-integrated instead.*

Yes, of course! I even did not think about flow-line approaches because the mass balance perpendicular to the flow line is crucial in the examples I have seen so far.

*Line 26: "The still limited performance of two-dimensional models arises from a combination of assuming hydrostatic pressure and the explicit time-stepping scheme implemented in almost all contemporary models (Bueler, 2023)." Like before, I would suggest using depth-averaged or integrated models, instead. Further I would ask to explain what performance is referred to? Pure computational performance or beyond that also in terms of accuracy and stability? In view of the latter (which links to my primary point of criticism), I would drop the "still", as to me it implies that the instabilities/inaccuracies described in this manuscript are something that could be overcome. In my opinion, they are built-in, as models based on hydrostatic approximation are not capable to resolve horizontal gradients in stresses that act on scales smaller than a few multiples of the ice thickness and assume small slopes (Hindmarsh, 2004).*

Performance is numerical performance here for me, but this performance is dominated by the instability of the usual schemes. So not accuracy in the context of narrow valleys, steep slopes or strong sliding etc.

The other point reflects our main disagreement. I agree that the hydrostatic approximation cannot resolve horizontal gradients in stresses that act on scales smaller than a few multiples of the ice thickness. However, as explained in the general part, I strongly disagree to the opinion that the numerical issues are an inherent property of the SIA and thus cannot be fixed by an improved numerical scheme.

*Line 62, eq. (3) and line 68 and eq. (5): "The factor involving the cosine of the slope angle $\beta$ of the ice surface with ... is a simple correction for steep slopes." I will come back to this later also in the appendix, but to me it appears that introducing this slope correction violates a principal assumption of the SIA that the surface normal in lowest order points exactly into the vertical direction which by shallowness also would demand that $\cos \beta \approx 1 \rightarrow \beta \approx 0$. In the other extreme, for $\beta \rightarrow \pi/2$ the 5th and 8th power of $\cos \beta$ in (9) will result in a vastly vanishing pseudo diffusivity and (8) basically reduces to the geodetic mass balance. In other words, mass transport shuts down on steep slopes. That in my view is artificial and covers the fact that the hydrostatic pressure approximation is not valid on slopes significantly deviating from zero (Hindmarsh, 2004) – see point one in major critics.*

Your argument with $\beta \rightarrow \pi/2$ is technically correct for a given vertical thickness. Here it happens because the thickness perpendicular to the bed goes to zero. This points towards a major disagreement (see general points and below): You say that the vertical thickness is responsible for the flow velocity, while I say that it is the thickness perpendicular to the surface. For discussing the effect of the slope correction, however, given vertical thickness is not the right point of view. It would rather be a given flux since the ice from the upper parts has to be transported. At given flux, the vertical thickness is then proportional to $(\cos \beta)^{-8/5}$, where $(\cos \beta)^{-3/5}$ comes from $\sin \beta$ instead of $\tan \beta$ and $(\cos \beta)^{-1}$ from converting vertical to surface-normal thickness. So there is no technical problem with the slope correction, and the question whether the SIA can be written in a rotated coordinate system is discussed below.

*Line 67: "The factor $f_d$ mainly depends on temperature in reality."* This factor should also contain some contributions from the vertical integral of the force balance, including the temperature dependence and enhancement factors. As demanded before, I think for completeness, the exact composition of $f_d$ as well as $f_s$ leading to the constant values reported in the text should be written out and explained – if not in the main part, at least in the Appendix. Further, as this seems to be the only mentioning of temperature, which, by the Arrhenius factor (Greve and Blatter, 2009) has a strong influence on the ice viscosity and hence the flow: I would kindly ask how the model – if at all – treats temperature variations in the ice, in particular as I would think that during a glaciation of the Alps the thermal ice conditions significantly vary, both, spatially and in time? Would the presented model be in principle capable of including advective temperature transport?

*Line 70: "The correction is explained in the derivation of Eq. (3) in Appendix A. It should, however, be mentioned that it is less elaborate than the correction for rapid mass movements developed by Savage and Hutter (1989) and only exact if the ice surface is parallel to the bed. In particular, it does not capture the effect of steep walls in a valley."* I discuss this in detail later where the method is explained in the Appendix. As mentioned before, I think the cosine in (3) is in violation of the basic assumptions behind SIA. I also do not see the parallel slab problem of a creeping shear thinning fluid in a direct connection to the Savage-Hutter theory, which is an expansion of the depth-averaged Navier–Stokes equations (i.e., including acceleration terms) for yielding granular flows, neglecting terms of order $O(\epsilon^{3/2})$ of the aspect ratio $\epsilon$, and in contrary to the SIA, usually already defined in a locally rotated coordinate system (e.g., Zwinger et al. , 2003).

Of course, I can replace the proportionality in Eq. A2 by the full expression

$$\frac{\partial}{\partial z} v_d(z) = A\tau(z)^n$$

with the ice deformation parameter $A$ ($\mathrm{Pa}^{-n}\mathrm{s}^{-1}$). Then we will arrive at the expression for $f_d$ as a function of $A$ and $\rho$ under the condition that $A$ and $\rho$ are vertically constant. In this case, the exponential Arrhenius factor would just occur as a factor in $f_d$. This would perhaps also help to become clear how to compute $f_d$ if $A$ depends on $z$. For $f_s$, however, this would not make so much sense because the sliding stuff is so uncertain.

Concerning temperature etc., I think that our interests differ fundamentally. From your previous work, I guess that you are interested in realistic, physically consistent, and comprehensive models. I am more interested in innovative solutions for individual model components.

Advective heat transport could be included easily as long as we are not interested in the vertical temperature profile because we could directly derive the energy flux from the ice flux. Depending on the scheme, we would just have to reduce $\delta t$ not to exceed the CFL criterion. And I would add a dispersion term to take into account the vertical velocity profile. If we include the vertical temperature profile, it would become technically more complicated and I am not sure whether I would do this with such a simple model.

I explained briefly in the general part why I think that key point of the Savage–Hutter theory can be transferred to the SIA and is consistent with the parallel slab solution. I try to provide some more details below.

*Line 80: "It should, however, be emphasized that modeling sliding along the bed is still one of the major challenges in this field. It is even questionable whether this process can be described well by the SIA or whether lateral stress components have to be taken into account." Please, refer to literature where these topics are addressed. For the latter: It is established (see literature cited in the major points section) that SIA is unsuited for fast sliding conditions.*

Of course, I do not mind including the references.

*Line 89: "Formally, it looks like a nonlinear diffusion equation, although it is mathematically not a diffusion equation due to the occurrence of $\nabla s$ in $D$. Can you please elaborate: If not a diffusion equation, what is it then?*

Not for the paper: It is still a parabolic PDE in the part of the domain where $\nabla s \neq 0$ because its linearized form is parabolic. Mathematical studies on nonlinear diffusion, however, typically do not allow $D$ depending on $\nabla s$. The reason is that the linearized form has a different diffusivity of $\tilde{D} = \frac{\partial}{\partial |\nabla s|}(D|\nabla s|)$, which would be $\tilde{D} = 3D$ here. Then all properties (diffusion of small disturbances, Fourier criterion, . . . ) depend on $\tilde{D}$ instead of $D$.

*Line 120: "The upstream scheme ensures $D = 0$ in this situation and thus avoids a permanent systematic increase in ice volume." What about the considerations on the second term of the r.h.s. of (8) at the contact line in order to avoid negative heights? There is often negative net mass balance, $r$, at the front of the glacier. Is it masked out? Generally speaking, to stay consistent in mass balance one should solve the mass balance or in higher order models the kinematic boundary condition together with the inequality for positive or zero flow-heights (see, e.g., Gagliardini et al. 2013, section 6.5), which in my opinion should be applicable to the scheme applied in this manuscript.*

There is no minimum thickness $h_{\min}$ implemented here, but it could be. And the workflow is:

1. Compute $D$

2. Compute $r$ and update the ice surface

3. Solve the ice flow equation

4. Remove negative thickness values

Steps 2 and 3 could be interchanged because the gradient in $r$ is typically small. If the thickness is negative and lower than $r\delta t$ (so more ice lost than melting), it is considered an error in the mass balance. At the front, however, this does not happen.

*Line 128: "Therefore, a dynamic smoothing of the slope of the ice surface in the diffusivity $D(h, |\nabla s|)$ is the central novel idea in MinSIA. The idea was originally motivated by Airy's linear theory of ocean waves (gravity waves in an inviscid fluid with a free surface). According to this theory, oscillations in pressure and particle velocity decrease exponentially with depth below the surface. The respective depth of penetration is proportional to the wavelength. Based on this result, MinSIA uses a smoothed version of $|\nabla s|$ with the length scale of smoothing being proportional to the ice thickness h." I am not able to see a connection from a theory based on gravity waves in Newtonian fluids to the mechanical behaviour of a (strongly non-linear and highly viscous) shear thinning free-surface thin-film Stokes problem. My theory: this smoothing algorithm works because it re-introduces the basic principle of thin-film approximations to resolve horizontal gradients with respect to a large aspect ratio (i.e., over several ice-thicknesses), even if run on mesh resolutions that apparently violate this assumption. To probe that theory, I suggest to run the simulation on a to the SIA appropriate mesh resolution ($> 1000$ m range) without smoothing and report on instabilities of such a run. I would also ask to include a clear motivation to opt for such small horizontal resolutions of 25 m–100 m in connection with SIA. Thinking of the main motivation presented in the text, a fast solution, I would even say that increasing the resolution should work exactly in this direction as the problem size gets smaller. This links to the first in the main point of critics.*

Since both reviewers do not like the motivation of the approach from Airy's theory, I can rewrite this part.

Technically, I agree to your theory. If we use a sufficiently large grid spacing, smoothing vanishes and even explicit schemes become stable. So there is no need to probe your theory technically. As discussed in the general part, our disagreement is whether the numerical instability is an inherent property of the SIA, whether the SIA becomes increasingly wrong just because we use finer grids and whether a coarse resolution that does not capture the valleys is better.

*Line 163: "The preconditioned conjugate gradients (PCG) method is used for solving the linear equation system. An incomplete Cholesky factorization is used as a preconditioner, whereby the version that compensates dropped nondiagonal elements at the respective diagonal elements turned out to be particularly suitable. A direct solver is also implemented for testing." As mentioned in the second point of major critics. I would ask for a more detailed explanation of this particular choice. What method was used for the direct solver? Were these methods utilizing specific features of the hardware (shared-memory or distributed memory parallelism, vector units, accelerators, if present)? What can the reader expect if this model is run on other, more modern platforms than the CPU that was reported in the text? Are there special license conditions to utilize the mentioned methods in MAT-LAB?*

The linear solver is not an essential part of the scheme and it should be easy to replace it by any other solver if desired. I am even not sure which direct solver MATLAB uses by default, but the direct solver is only interesting for small test examples since it requires much memory. The preconditioner was found by performing experiments with several different preconditioners and settings, but already knowing that incomplete Cholesky factorizations are not too bad for problems of this type. The C code for the entire iterative solver in the Python implementation will, of course, be open and consists of less than 100 lines.

But to be honest, I am not interested so much in the hardware-related aspects. There were no parallel-computing features used explicitly, which means that MATLAB just uses multiple cores for builtin functions. Practically, however, it was not much more than 1.5 cores on average (read from the Linux top). In principle, everything except for the steps related to preconditioning could be parallelized. So there might be a scale at which the preconditioner becomes the bottleneck, so that another preconditioner might become better.

*Line 187: "As a first test, the accuracy of the solutions for different values of the smoothing factor $f$ with respect to the reference solution ($f = 0.25$, $\delta t = \frac{1}{64}$ yr) is measured." In relation to the major point of critics, I do not think that comparison to a result obtained with the method is able to show the accuracy of the method itself. To me, it merely shows the sensitivity of the method to a change in numerical parameters and resolutions.*

Since the first reviewer also asked for a simulation without smoothing, I started one. As stated in the manuscript (and also in your next comment), it is in principle too expensive. So I started it from $t = 2900$ yr and it made 60 yr so far. For $\delta t = 1/2048$ yr at $\delta x = 25$ m, there are still considerable staircase oscillations, which vanish for $\delta t = 1/4096$ yr. The maximum oscillation (over all nodes of the grid) rapidly falls below $10^{-4}$ m and stays there.

A bit more analysis: Revisiting Fig. 2(b), we can see that the deviation between $f = 1$ and $f = 0.5$ is about twice as big as the deviation between $f = 0.5$ and $f = 0.25$. This finding tentatively suggests that the error resulting from smoothing is linear in $f$. In this case, the deviation between $f = 0.25$ and $f = 0$ should be as big as the deviation between $f = 0.5$ and $f = 0.25$ (from a geometric series $0.5 + 0.25 + 0.125 + \ldots = 1$). The numerically obtained deviation between $f = 0.25$ and $f = 0$ at $t = 2960$ yr (so after the 60 yr) is even lower than this, but this may be due to the short time span. Nevertheless, the deviations between $f = 0.25$ and $f = 0$ seem to be in the order of magnitude of a few centimeters, which seems to be sufficiently small to me to consider $f = 0.25$ as a reference.

*Line 209: "Since even explicit schemes are stable for sufficiently small time increments, the occurrence of the staircase oscillation must be related to both the smoothing factor $f$ and the time increment $\delta t$." I would like to see a proof on the statement of stability for small timesteps. Either by citing literature or running the model (at least for some time) with no smoothing on the 25 m resolution. What I think, is that the oscillations are a reaction of the SIA to significant undulations with a length-scale way below the ice-thickness. Consequently, I would conclude that the oscillations are natural to the SIA (if run on resolutions that violate the shallowness assumption) and damped out by the smoothing, similar to as they would by running the SIA without smoothing on coarser meshes (which I already asked you to test) that resolve horizontal scales above the typical ice-thicknesses.*

The point that the oscillations arise from undulations with a length-scale way below the ice-thickness is correct (the oscillations with the worst impact are on the scale of $\delta x$), but the rest of your statement reflects our main disagreement. As stated in the general response, the SIA is practically a diffusion equation (the fact that the linearized diffusivity is $3D$ instead of $D$ does not matter here), which means that these undulations do not result in growing oscillations for an exact solution. Exact solutions of diffusion-type equations smooth such undulations without overshooting. The exact solution dampens such undulations even faster if the wavelength is shorter. The conversion of surface undulations into oscillations is fully a matter of the numerical scheme. This is confirmed by the simulation without smoothing and $\delta t = 1/4096$ yr reported in the previous point.

*Line 267: "The simulations over the entire time span with $\delta t = 0.25$ yr took $35400$ s for $\delta x = 25$ m, $6740$ s for $\delta x = 50$ m, and $1290$ s for $\delta x = 100$ m on an Intel Core i5-7600 CPU (3.50 GHz) from 2018." This, to me, seems to be the only information on how some sort of performance evaluation has been done. In my view, this is not enough such that the reader can get a clear picture what to expect if running the model themselves. I would ask to specify the utilization of the CPU (single core or multiple cores, as there seem to be 4 included in the reported model). If multiple cores have been utilized, please report on what parallel paradigm was deployed (shared or distributed memory). How much memory was available in the test platform and how was the saturation of the available memory at the runtime? This links to the second point of my major critics.*

Yes, this is the only information, and I feel unable to provide much more information. It should be immediately clear that it is much faster than Elmer/Ice and PISM, but not a comprehensive model including heat transport etc. and with the limitations of the SIA. From my experience, it is also much faster than the implementation of the iSOSIA in the respective erosion model. And a comparison with the AI-based IGM seems to be extremely difficult to me at the moment (usage of GPU computing, time for training, . . . ). I can finally add the times for the Python version.

*Line 288: "Figure 8 illustrates the weakness of the scheme at steep slopes for $\delta x = 25$ m and $\delta t = 0.25$ yr." I would rather say it reveals the inapplicability of the approximation than a weakness in the numerical scheme (see major point of critics).*

Definitely not at this occasion. As written in the text, this applies to steep slopes with rather small ice thickness. Here the shallow-ice or slab approximation itself is ok. It is just that advection dominates over diffusion at these points and that the scheme is not better than any other scheme then.

*Line 305: "At present, a MATLAB implementation of MinSIA is available under the GNU General Public License. A Python implementation is under development." As mentioned under the major points and before in this section – despite the setup being shared with an open license – the proprietary nature of the software needed to run it imposes a big hurdle for people to reproduce these findings. This all would have been no problem if the publication would have been postponed to the point in time when the announced Python version would have been available.*

I hope that is point was addressed sufficiently in the response to an earlier comment. In this specific case, I could imagine that you would indeed have tested it without smoothing and a small $\delta t$. But I honestly do not know many reviewers who would have spent this work. And would if have had any effect on your opinion that everything that I wrote about partial differential equations and about the SIA is wrong?

*Line 334: "A 2-element array corresponds to $r_{\max} = \infty$ and a scalar value additionally to $g_- = g_+$." I am not able to understand the meaning behind this sentence. Can you please elaborate? Do you allow for infinitely large accumulation rates?*

It just means that you can omit he third entry of the array if you want unlimited ice production and that you can omit the second entry if you want the ablation gradient to be equal to the accumulation gradient. May be useful or not, but it is implemented like this.

*Line 390: "Let us, for simplicity, assume that the ice surface is parallel to the bed with a slope angle $\beta$ and that the $z$-coordinate is perpendicular to the surface with $z = 0$ at the bed." I think this is the culprit of the discrepancy about the bed-slope correction I raised in the main issues. The parallel slab example (see Greve and Blatter, 2009), which seems to be adapted here, in my view, cannot be intermixed with the derivation of the SIA. The basic assumption behind the SIA is a directly into the vertical direction pointing surface normal (Greve and Blatter, 2009). The resulting full alignment with the negative direction of the gravity is a necessary condition for the zero Cauchy stress vector at the surface leading to identically vanishing stress-components and pressure, which constitutes another necessary condition for the vertical integration of the shear stresses (Greve and Blatter, 2009) in SIA. If the integration in the vertical direction is done from bedrock, $b$, to the free surface $h$, one obtains the shear stress at the bedrock as $\tau = \rho\rho gh\nabla s$, where $h = s - b$, i.e., the total ice-depth in vertical direction. Thus, in SIA the shear stress is imposed by the hydrostatic pressure gradient induced by the horizontal change in surface elevation (with no influence of the bedrock slope at all), whereas in the parallel slab, i.e., equation (A1), the shear stress is the result of the downslope weight component of a in the rotated reference frame bed- parallel surface with in this rotated reference frame identically vanishing gradients of the free surface. As I see it, the author is setting (A1) identical to the bed-shear stress as defined in SIA, which I claim violates the basic assumptions of the SIA. If the author disagrees, I would ask to include a detailed derivation of the SIA (starting from the Stokes equations) in a rotated reference frame and show that the derived equations still comply with the lowest order approximation, even for large bedrock angles, as they occur in the studied topographies of the Alps and even the Black Forest.*

For simplicity, linear Stokes equation with constant viscosity,

$$-\nabla p + \rho\vec{g} + \eta\Delta\vec{v} = 0,$$

and the same direction of the velocity everywhere,

$$\vec{v} = f(x, y, z)\vec{e},$$

with a (constant) unit vector $\vec{e}$. Then

$$\Delta v = (\Delta f)\vec{e}.$$

Define a normal vector $\vec{n}$ perpendicular to the velocity, so $\vec{e} \cdot \vec{n} = 0$ and obtain

$$\nabla p \cdot \vec{n} = \rho(\vec{g} \cdot \vec{n}),$$

which means that the pressure gradient perpendicular to the velocity is the component of gravity perpendicular to the velocity. I leave the rest of the derivation, including the generalization to the nonlinear flow law, to you. But for sure: hydrostatic perpendicular to the velocity and not necessarily in the vertical direction.

For completeness, the hydrostatic pressure at the bed (valid for Stokes and Navier–Stokes) in Cartesian coordinates (see, e.g., Hergarten 2024, doi 10.5194/gmd-17-781-2024) is

$$p_b = \frac{\rho gh}{1 + \nabla s \cdot \nabla b}.$$

Your SIA version ($p_b = \rho gh$) is obtained for $\nabla b = 0$ and the slab version (my correction terms) for $\nabla b = \nabla s$, which leads to

$$p_b = \rho gh \cos^2 \beta.$$

Just to mention: Hutter (1983, 0.1007/978-94-015-1167-4) defined $\nabla b \approx \nabla s$ as a condition.

---

## Author Response (AR1)

Dear editor, dear reviewers,

thank you for considering my manuscript! First, I would like to point out that the manuscript was never intended to overcome the physical limitations of the SIA. It only presents a highly efficient numerical scheme that allows for spatial resolutions that cannot be reached by any other model at a reasonable computing effort. Let me briefly summarize the options for numerical ice-flow simulations.

1. Full Stokes (3-D) simulations, e.g., based on Thomas Zwingers' Elmer/Ice approach are the best option concerning the underlying physics, but after 20 years still limited to a small group of researchers due to the high computational effort even at moderate spatial resolutions.

2. Extensions of the SIA by horizontal stress components are in some cases not much behind full Stokes, but in turn computationally not so much more efficient than full Stokes as one might expect.

3. AI-based approaches have brought life into the field. These approaches achieve "reasonable" computing times on GPUs instead of high-performance clusters and already have motivated several of my colleagues to start with ice-flow simulations. However, several researchers still feel a bit uncomfortable concerning AI.

4. The new numerical scheme for the SIA is competitive to the AI-based approaches concerning the numerical efficiency (perhaps even on desktop PCs instead of GPUs), but suffers from the inherent limitations of the SIA. Most important for alpine topographies, flow velocities are overestimated around the center of narrow valleys and vice versa. The new numerical scheme can, of course, not overcome these physical limitations.

It was not clear to me that physical limitations of the SIA are such a big story for the community. Some of the comments even read as if the SIA became increasingly wrong with finer resolution, which does not make sense. There is the numerical approximation error, which decreases with finer resolution, and the physical error from neglecting horizontal stress components, which is basically independent of $\delta x$ and therefore becomes dominant for $\delta x \to 0$.

The main point of reviewer 2 is that high resolutions do not make sense then. I am coming more from the field of rapid mass movements, where the deficiency of the Savage-Hutter approach is basically the same. There it has been discussed a bit (Hutter et al. 2005, doi 10.1098/rsta.2005.1594), but nobody really cares since 3-D models are still in their infancy.

My idea was to propose the numerical scheme and a minimalistic implementation, which might enable other researchers to extend the scheme for better approaches that include horizontal stress components. For the existing approaches (either higher order or SIA/SSA combinations), it seems not to be trivial to maintain the extremely high numerical efficiency. Maybe I will end up with a new, simpler approximation. This will, however, have to be tested extensively and will take some time.

A rejected preprint could, of course, also be the basis of further development. However, I feel that I already spent too much time on the comments of the reviewers to give up without submitting a revised version.

Best regards,
Stefan Hergarten

**Editor (Ludovic Räss)**

*In particular, the revised version should make sure to clearly stress the benefits and limitations of the presented approach, and make sure to perform all relevant benchmarks in the regime or configuration that is most valid for the SIA framework.*

I think that it is not a matter of stressing the benefits. The only benefit is that we can formally reach resolutions that we need for alpine topographies, which has not been possible with any model before. **Concerning the limitations, I added a section to illustrate the issue with the velocities in narrow valleys and to point out the difference between the numerical approximation error and the physical error (Sect. 6).** However, "having the benchmark in the regime or configuration that is most valid for the SIA framework" is not so easy. If I can convince reviewer 2 that the extension for inclined bedrock surfaces is correct, this configuration would be an inclined plane. Otherwise, it would be a horizontal plane. In both cases, it would be difficult to explain what the approach is good for.

*Also, note that the high-resolution could also be formulated in grid points rather than in meters, which could further highlight that being able to solve SIA with many DoFs is not only useful when aiming at small grid spacing but would also allow to investigate larger domains.*

A good idea in principle, but the problem is that time scale and spatial scale are connected in diffusion-type models. As classical explicit schemes suffer from a quite strong limitation of the maximum $\delta t \propto \delta x^2$, the advantage of implicit schemes fades for large $\delta x$. Even if I was able to simulate ice sheets with $\delta t = 10,000$ yr instead of 10 yr at a few kilometers spatial resolution, it would be difficult to explain what is is good for because the conditions do not stay constant over such scales. It would have been possible to simulate glaciations of the Alps with $\delta x = 200$ m, but the criticism about the physical limitations of the SIA would have been the same.

*Regarding convergence, SIA is a PDE and as such it should be solved with mesh convergence. Specifically, it would be relevant to provide mesh convergence experiment and report about the findings during revision. Also, running SIA on resolved vs filtered coarse topography may not show much discrepancy in terms of resolving physics, but may be undeniably better in terms of achieving converged results.*

My first draft included such an analysis, but I found it not clear and important enough at that time. **I included some results (Fig. 8), which suggest that the theoretical linear convergence of the upstream scheme is reproduced quite well at least at $t = 300$ yr.** At $t = 3000$ yr, the convergence is weaker than linear. I think that this is not a matter of filtering the topography, at least not on the scale of $\delta x$. It is rather the formation of almost horizontal ice surfaces, which generate a gradient in thickness opposite to the bedrock gradient. Since the dependence of the diffusivity is highly nonlinear, it seems to be the averaging over the diffusivity that makes the convergence slower than predicted by the upstream scheme. So the slower convergence rather seems to arise from the existence of valleys than from the small-scale roughness.

*Finally, if the authors are not willing to provide a (less performant) Python version to support open-source code usage, I would remove the related bits from the manuscript's abstract.*

Any attempts with the standard SciPy CG solver and different preconditioners were really poor concerning performance. I finally ended up with a short C++ code for the PCG scheme and the preconditioner (about 100 lines in total), which has to be included as a precompiled .so (Linux, MacOS) or .dll (Windows). While the C++ code must be compiled by the user, a precompiled .dll for Windows is included in the repository. The Python version was tested extensively with several of the benchmark scenarios and the numerical efficiency is surprisingly good (even 1.5 times as fast as the MATLAB version).

**Reviewer 1 (Daniel Moreno-Parada)**

*The current work explores the numerical benefits of a new algorithm to solve the Shallow Ice Approximation (SIA). It combines a semi-implicit time-stepping with a dynamic smoothing to deal with the nonlinear flow velocity. The scheme is only implemented in MATLAB, as the Python version is still under development. Numerical tests are performed to study the numerical stability in terms of the smoothing factor and timestep, focusing on the potential staircase oscillations and mass conservation.*

*Overall, I find that the message of the paper is successfully conveyed. Nevertheless, I have a number of major remarks that I would like the author to address and further elaborate on the manuscript. In particular, the work falls short in certain aspects that I have listed in the Major Remarks section. Lastly, I have included a short list with other minor remarks at the end of the document.*

I was happy to read that the reviewer found that the message of the paper is successfully conveyed, but feel that he did not get some points that are important for me completely. I gave my best to make these points clearer in the revised version.

**Analogy with Airy's linear theory.** *The author did not provide any physical justification of the analogy with Airy's wave theory: a linearised description of the propagation of gravity waves on the surface of a homogeneous fluid layer. A priori, this problem fundamentally differs from the SIA description of glacier ice tackled by MinSIA. In my view, there are several flaws:*

- *Airy's exponential decay is a consequence of water wave kinematics (Dingemans, 1997), not directly transferable to ice sheet mechanics. How is the horizontal smoothing in ice surface justified by the exponential decay along the vertical dimension in Airy's linear theory? Why larger ice thicknesses imply a larger length scale of smoothing?*

- *Ice behaves as a non-newtonian fluid, not considered in Airy's theory. Viscous dissipation in ice dampens short-wavelength perturbations, altering how surface effects propagate.*

*In summary, the analogy lacks physical justification. Ice thickness does not play the same role as wavelength in water waves decay, unless otherwise shown by the author. With the current description, the smoothing appears as an heuristic fix (as stated by the author: "the thickness-weighted average avoids artifacts at the boundaries of ice-covered area"), not a physics-derived approach. I encourage the author to either provide a detailed physical justification of the analogy or remove the reference to water wave propagation, thus re-framing the motivation.*

**I removed it and extended the numerical reasoning a bit (lines 166–174).** However, I did not claim that the smoothing approach can be formally derived from Airy's theory. It was only used as an example that short-wavelength oscillations do not accelerate the entire column of material as assumed in the shallow water/ice approximation, but typically have a limited depth of penetration.

While I agree that the idea cannot be transferred quantitatively to a flow law with nonlinear friction, let me briefly explain why it is not as weird as it may seem: If $h$ is thickness and $d$ depth of penetration, average acceleration of the entire column is

$$a \propto \frac{1}{h} \int_0^h e^{-\frac{z}{d}} dz = \frac{d}{h} \left( 1 - e^{-\frac{h}{d}} \right).$$

So $a \propto d \propto$ wavelength for $d \ll h$ and $a = $ const for $d \gg h$. The effect of surface oscillations on the entire column increases with increasing wavelength at first, but then flattens. Practically, $d$ ($\propto$ wavelength) defines the wavelength at which the effect does not increase much further. So the idea was to cancel all oscillations with a wavelength shorter than the wavelength defined by $d$.

**Inadequacy of SIA.** *As described by Greve and Blatter (2004), the SIA is justified for large ice sheets where conditions generally vary little over horizontal distances (i.e., 5–10 times the local ice thickness). However, this is not often the case in mountain glaciers, where longitudinal and transverse coupling of stresses are important and should therefore be considered. To illustrate this inadequacy, Fig. 7.4 (Greve and Blatter, 2004) shows a scatter plot comparison of the First Order Approximation and the SIA velocities for Haut Glacier d'Arolla: the latter underestimates the velocities for values below 10 m/yr, and vice versa for velocities above this threshold.*

*Given that "The purpose of this paper is to challenge the hypothesis that the potential for further improvements in computational efficiency with classical numerical methods is limited (Jouvet et al., 2021)", I encourage the author to choose a more suitable stress balance approximation and discuss the differences with a more sophisticated description. If the SIA is nonetheless correct, then a simple plot as Fig 7.4 in Greve and Blatter (2004) will suffice to justify and discuss the choice over the domain. Note that the work of Jouvet et al, (2021) employed PISM output to train the emulator, capturing the physics contained in longitudinal stresses present in the SSA. It would be also convenient to discuss how the second-order SIA performs in this context (e.g., Ahlkrona et al, 2013.)*

It was not my intention to convince anyone to use the SIA. The numerical efficiency is really high (much higher than that of the classical non-AI approaches and than the reviewer probably realizes), but this efficiency has its price as long as there is not extension by horizontal stress components available. Admittedly, some researchers who already started to use MinSIA do not care much about the limitations of the SIA. **So I added a short section about the limitations of the SIA for readers who are not so aware of these limitations (Sect. 6) with a section across one of the "worst" valleys (Fig. 11) and a profile where it should be not so bad (Fig. 12).**

I am still hoping that anyone will extend the scheme for a higher-order approach or for the SIA/SSA combination, although it is not trivial to maintain the numerical efficiency. I also started working on an extension by horizontal stress components, which will be even a bit simpler than the SIA/SSA combination, but only take into account horizontal shear stresses. This extension will be based on a second partial differential equation for the diffusivity. But since this will a new approximation, it will have to be tested seriously against full Stokes and/or iSOSIA. Given that I cannot work on this topic for the whole day, implementing and testing will probably take one year.

**Lack of smooth-free reference.** *As stated by the author: "It would be desirable to use a simulation without smoothing as a reference scenario". There is an inevitably trade-off between smoothness and numerical stability. I consider that it is thus mandatory to have a reference simulation to quantify the deviation from the smooth-free solution. As stated in the paper: "the smoothing factor was set to $f = 0.25$ as a reference". Why is it so? This number seems arbitrary without further justification. In fact, it corresponds to a strongly oscillating result (Fig. 4) that need a nearly 108 m3 of ice in order to keep the ice surface consistent (Fig. 5). Please, provide detailed physical justification on why $f = 0.25$ should be a reference value.*

I do not understand how the "strongly oscillating result" was derived from Fig. 4 and "need a nearly $10^8$ m$^3$ of ice in order to keep the ice surface consistent" from Fig. 5. For $f = 0.25$ and $dt = 1/64$ yr (the reference), the 95 % oscillation (exceeded in 5 % of all time steps) is about $2 \times 10^{-5}$ m and the absolute maximum oscillation about $5 \times 10^{-4}$ m. Given all the uncertainties in glacier modeling, I would not consider oscillations of less than 1 millimeter strong. The added volume is always less than $3.5 \times 10^{-5}$ m$^3$ in each time step and less than 5 m$^3$ added over the entire 3000 yr simulation, which seems to be small in relation to the total ice volume of about $5 \times 10^{11}$ m$^3$.

**I added some results from a shorter simulation (from $t = 2900$ yr to $t = 3000$ yr with $\delta t = \frac{1}{4096}$ yr) to illustrate that the difference between smooth-free ($f = 0$) and reference ($f = 0.25$) is clearly smaller than the difference between $f = 0.25$ and $f = 0.5$ (lines 257–264 and new Fig. 4)**.

**Python implementation.** *As stated by the Editor, MATLAB is not the most "open software", so it would be convenient to provide the Python software under development and discuss the performance in the manuscript. In this line, NVIDIA recently announced the cuNumeric library, a drop-in replacement for the NumPy library that allows to run on multi-core CPUs, single or multi-GPU nodes, and even multi-node clusters without changing your Python code. Operations are executed by Legate's task engine and accelerated on one or many NVIDIA GPUs (if no GPU is present, on all CPU cores). Linking with the previous comment, as the author justify the absence of smooth-free reference simulations as a results of prohibiting computational costs, this apparent issue could be potentially overcome by using cuNumeric library. Either way, since the main focus of the paper is to show that there is still room for improvement in computational efficiency with classical numerical methods, I encourage the author to include a section where parallel performance is elaborated.*

Thanks for the suggestion and sorry for not making clear that the problem with the efficiency of the Python version will be solved in a reasonable time. Looks like a good suggestion in general, but it would still be bad without an NVIDIA card. Any attempts with the standard SciPy PCG solver and different preconditioners were really poor concerning performance. I finally ended up with a short C++ code for the PCG scheme and the preconditioner, which has to be included as a precompiled .so or .dll. With this precompiled solver, the Python version it is even about 1.5 times as fast as the MATLAB version. And since it is already about 2–3 orders of magnitude faster than an explicit scheme, I will not evaluate the parallel-computing performance at the moment.

**Missing comparison with measured glacier velocities.** *I consider that the perturbation induced by smoothing should be rather framed in the context of observed glacier velocities. It is expected that timestep and spatial resolution will impact the presence of numerical artefacts such as the oscillations discussed, but a comparison with observed velocities is fundamental. If this paper aims at keeping "classical numerics competitive", it requires some sort of validation with measured velocities. My suggestion would be to compare Min-SIA results with observed values as Jouvet et al. (2021). This will serve as a validation test to quantify to what extent smoothing perturbs the velocity field.*

**I hope that it becomes clearer with the extended explanations (Fig. 4 and Sect. 6) that there is really nothing beyond the inherent limitations of the SIA.** The perturbation induced by smoothing is negligible for $f \leq 1$. Without smoothing, there are considerable oscillations at the surface due to the staircase-like ice surface. Of course, these oscillations would cause unrealistic velocities. However, these oscillations vanish already with moderate smoothing. There are really no artifacts except for the physical limitations of the SIA.

**Overstated stability claim.** *Line 107 of the manuscript reads: "This semi-implicit scheme already ensures stability for arbitrary time increments $\delta t$". This claim is quite strong and generally incorrect for this specific scheme. If $D$ changes rapidly in space or time, evaluating it explicitly can lead to instabilities for large $\delta t$. The scheme is not unconditionally stable. In fact, the author later states in the paper that special focus is needed on the timestep to avoid numerical oscillations (see my next comment) and even discusses situations where the CFL criterion is relevant. Please, revise other vague statements regarding numerical stability.*

For this discussion, it would be helpful to know which definition of instability the reviewer is referring to. Although there is formally no unique definition, all definitions I am aware of that can be applied to diffusion-like equations are similar in their spirit and lead to the same result. As far as I know, instability is always related to an exponential growth of small disturbances. Here, the simplest definition would be that the variation in the solution (here maximum $s$ minus minimum $s$ over the domain) must not increase (as it is the case for exact solutions). **I added a short sketch of a proof of this unconditional stability (lines 136–141) and some explanation to clarify the difference between stability and accuracy (lines 157–165).** However, I would like to point out that there were no "vague statements regarding numerical stability".

**Focus on "time increment"** $\delta t$**.** *The paper reads that: "[dots] the systematic error in the volumetric balance is negligible compared to the immediate effect of the staircase oscillations on accuracy. So focus should be on limiting $\delta t$ to avoid these oscillations". Can we simply conclude that, given the numerical nature of a semi-implicit scheme, the stability is determined by the timestep? (and therefore contradicts the previous claim that the semi-implicit scheme ensures stability for arbitrary timesteps). If so, a fully implicit scheme would overcome this issue? Further experiments are needed to support or reject this hypothesis.*

This point is closely related to the previous point about stability and accuracy. We cannot conclude that the stability is determined by the timestep (it is unconditionally stable according to the definitions I am aware of), but the accuracy depends on $\delta t$. And, specific to this problem, there is a quite sharp transition from reasonable to poor accuracy. This is why there is a quite strict limitation of $\delta t$. And yes, a fully implicit scheme would overcome this issue as far as I can see, although it is mathematically not trivial due to the occurrence of $|\nabla s|$ in the diffusivity. However, I am not aware of any fully implicit scheme that also converges for large $\delta t$. Each explicit scheme and each semi-implicit scheme can be extended to a fully implicit scheme, e.g., by a fixed-point iteration or by a Newton iteration. However, these schemes typically achieve a reasonable convergence only up to time increments in the same order of magnitude as the respective underlying explicit or semi-implicit scheme. Practically, these schemes are even less efficient than the the respective explicit or semi-implicit scheme.

As an example, Bueler (2016, doi 10.1017/jog.2015.3) proposed such a scheme, admitting later (Bueler, 2023, doi 10.1017/jog.2022.113) that "The scaling of the Bueler (2016) implicit SIA solver is not enough better than such an explicit scheme, however." The same holds for the predictor-corrector schemes promoted in the next point. As far as I know, there is no fully implicit scheme for the SIA that converges for arbitrary $\delta t$. So there is no basis of further experiments.

**Timestepping and redundant sections.** *I would suggest merging Section 4.2 (The maximum time increment) and 5 (Finding the best time increment) in a single section. All numerical results regarding timestepping should be described and discussed therein.*

*Moreover, I find a great amount of effort on the present work while lacking some important points. For instance, Cheng et al. (2017) introduced an adaptive time step control for simulations of the evolution of ice sheets using Elmer/Ice (Gagliardini et al., 2013). Semi-implicit and fully implicit methods are compared for a number of discretization stencils. I consider that if the problem of "finding the best time increment" is to be tackled, the paper should dive into the predictor-corrector (among others) approaches and showcase the performance in MinSIA.*

The first draft of the manuscript contained such a merged section, but then I decided to separate the practical aspects of finding a "good" $\delta t$ from the numerical analysis (and included some results from the Alpine examples). I still think that it is better to keep this more practical section for those who do not want to go though the entire numerical analysis.

However, I am afraid that we are still speaking about different orders of magnitude. For the high-resolution example with $\delta x = 25$ m (which both reviewers may find not useful), the maximum stable time increment is $\delta t = \frac{1}{32768}$ yr. If I read it correctly, the predictor-corrector schemes in the suggested reference (Cheng et al., 2017) do not allow for larger $\delta t$, but just yield a better accuracy for smaller $\delta t$. Similarly, the surface stabilization for Elmer/Ice recently proposed by Löfgren et al. (2024, doi 10.5194/tc-18-3453-2024) increases the maximum $\delta t$ by a factor of 2 to 5. I would not dare to say that this is a minor progress, but here I am proposing a scheme that allows for an increase by more than 3 orders of magnitude (up to $\delta t = 0.25$ yr at $\delta x = 25$ m).

*Figure 1. This plot is hard to interpret: colour bar is missing, legend is missing, spatial scale is missing, inset with geographical zoom-out is also missing. Please, improve the figure so that they are as much self-explicative as possible.*

**I added a color bar and a 5 km grid for the horizontal scale.** A simple scale bar is not possible in the 3-D representation since the scale in view direction is different from the scale perpendicular to the view direction. I do not like the grid, but it is ok for me if requested. However, it is not clear to me which information should be included in a legend and what a geographical zoom-out should be good for.

*It would be very convenient to include explicit discretization schemes implemented in MinSIA. Section 3 (Numerical scheme and implementation) elaborates on the smoothing algorithm, but the finite volume and upstream diffusivity schemes are not given in the manuscript. An appendix with the explicit discretization is beneficial for reproducibility and future comparison.*

I feel that it would be a bit trivial. Finite-volume diffusion on a regular grid should be clear for those who get so deep into the implementation. And then taking the diffusivity from the node with the higher surface elevation is also simpler in words than in mathematical symbols.

*I wonder how the CFL criterion looks like overlaid in Fig. 7. It is illustrative to show the timestep restriction imposed by the CFL criterion for different resolutions and smoothing values. Moreover, Fig. 7 should also include the deviation from the smooth-free reference simulation. Large values of the smoothing factor could imply unrealistic velocity fields.*

Since the velocities are almost independent of the smoothing factor (as long as the oscillations are not too strong), the CFL criterion would be a horizontal line. Roughly $\delta t = 1/20$ year for $\delta x = 25$ m, $\delta t = 1/10$ year for $\delta x = 50$ m, etc. Not sure whether this information provides new insights. The main problem would, however, be that how much I can exceed the CFL criterion seems to be problem-specific. I cannot make a general statement like "5 times higher than the CFL limit" at $\delta x = 25$ m.

And what is meant by the "deviation from the smooth-free reference simulation"? I could include a data point left down for with $f = 0$ (unfortunately, a bit difficult with the logarithmic scale), $\delta t = 1/4096$ yr and the respective values for the other $\delta x$, but I am not sure whether it improves the figure since the relevant information would be compressed into the upper right corner.

And again, there is nothing beyond the inherent physical limitations of the SIA. Smoothing only helps to keep the surface gradient at ice surfaces with little inclination constant. Strong artifacts in the velocity field would only arise if smoothing goes from one valley into another valley. As an example, the quite narrow valley shown in the new Fig. 11 has a thickness-to-width ratio of about 0.25. For $f = 1$, smoothing of the surface gradient would capture the entire width according to the definition of $f$ at the center line, but with decreasing weights outwards. For $f > 1$, it would start to include the slopes, but they have little effect due to the thickness-weighted averaging. Getting into the next large valley would require much larger values of $f$. This is the decrease in accuracy at large $f$ (2, 4, 8), but this is outside the interesting range.

*Line 5: What does this mean: "MinSIA is designed for simulations with several million nodes on standard desktop PCs"? What does the author mean by node here? Regular desktop PCs usually have only ∼ 16-32 CPU cores.*

Nodes of the finite-volume grid. **I added "on grids" (line 6).**

*Lines 264–270. This paragraph could be synthesized by plotting the computing time as a function of different parameters (e.g., $\delta t$, $\delta x$, $f$, etc.). In log-scale, the slope of the linear fit will show the exponential dependency.*

I thought about this, but I have a very limited amount of timing data available. Since the computations were performed on different PCs with different numbers of processes running, I cannot reconstruct the computing times from the metadata of the output files.

**Reviewer 2 (Thomas Zwinger)**

**General Impression**

*The submitted manuscript describes the implementation of an ice-flow model based on the Shallow Ice Approximation (SIA) formulated using the vertically integrated mass balance and implemented in MATLAB. The manuscript further discusses measures in spatial smoothing and time-discretization that had to be taken to stabilize the method in order to produce simulations using synthetic forcings of glaciations over selected regions of the Alps and the Black Forest, the latter used as the benchmark case presented in this study. Whereas the description of the model seems to be almost complete (I ask for a few more details) and implemented along the guidelines of the target journal (GMD), to which general purpose model descriptions fit, I have to express major concerns on the applicability of the model in view of its approximation and the applied slope correction and the evaluation of its performance and reproducibility by the reader, which I try to elaborate in the sections below.*

In short:

(1) Of course, the new numerical scheme cannot overcome the deficiencies of the SIA arising from the simplified physics. As already discussed in a second round, however, I do not fully agree about the consequences of these deficiencies.

(2) I completely disagree to the concerns about the slope correction (see below).

(3) The main concern about the reproducibility should have been fixed with the availability of the Python version.

**Critical issues**

*Looking at the abstract text, I already was surprised to read about 25 m spatial resolution (assuming it is the one in horizontal direction) in connection with the applied Shallow Ice Approximation (SIA). This seems to have been confirmed within the text. There is a long series of studies on the validity and accuracy of the SIA by several authors (Johannesson, 1992; Gudmundsson, 2003; Hindmarsh, 2004 – to list a few) that later were accompanied by numerical investigations comparing full-stress Stokes (e.g., Le Meur et al. 2004, Seddik et al. 2017 ) and/or higher order approximation models (Pattyn et al., 2008) to SIA. The quintessence of all these studies is that SIA is inaccurate with respect to three aspects that apply to the studies presented in this manuscript, namely, (1) fast sliding (though this is difficult to estimate speeds from the information given) (2) steep slopes and increased accumulation (3) terrain resolutions significantly below a few multiples of the local ice thickness. The last point, I would even say, is a validity criterion, as shallowness (which also applies to longitudinal gradients to be resolved) is the fundamental parameter behind the expansions of the Stokes equation leading to the zeroth-order equations, the SIA (Hindmarsh, 2004). In other words, based on the above mentioned literature, to me a resolution below about a few times the local ice-thickness is in violation with the principle assumption that go into the derivation of the approximation applied in this study. I am not surprised by the instabilities arising from resolving steep glaciers at 25 m with SIA.*

This point was already addressed to some degree in a second round of discussion. There is no doubt about the physical limitations of the SIA arising from neglecting all horizontal stress components. **I added some sentences about the limitations in the introduction (lines 22–24 and 27–30), a short section to explain the deficiencies of the SIA for a narrow valley, where the problem is worst (Sect. 6), and included the references.** However, we have to distinguish between physics (limited accuracy that cannot be overcome by mathematics or numerics), mathematics (still a well-defined partial differential equation), and numerics (instabilities, which are a deficiency of the existing numerical schemes). I also tried to clarify that the horizontal length scale that theoretically has to much larger than the ice thickness is not necessarily related to the spatial resolution of the numerical model. This means that the inaccuracy from the limited physics becomes dominant compared to the numerical error at high resolution, but this does not necessarily affect the entire domain and does not make high-resolution simulations wrong immediately. Concerning the "bad conditions", I would say that (1) – fast sliding – is correct in principle, but it is difficult to measure in field and it is not trivial that other approaches are so much better. I disagree concerning (2) – steep slopes – (see below). (3) is undeniable true, but not in terms of resolution, but terrain scales (in particular, the width of valleys).

*I would also like to see a mathematical argument to whether the ad-hoc introduced slope correction in equation (3) is consistent with the assumption of the SIA (I think it is not). The author leaves the reader with no means of quantifying the inaccuracies introduced by the choice of the method in combination with the resolution and steepness of the terrain, since the reference run is provided with the method itself. It either would take comparison with observations (not possible for synthetic ELA forcing) or a higher order (if not Stokes) solution to the problem. To me, these issues seem not to be fixable by improved numerical methods, as they are inherent to the approximation applied. Thus, I would see the necessity to demonstrate the mathematical consistency of the presented model with the zeroth order approximation (SIA).*

From the specific comments, I conclude that the crucial question is whether the pressure is vertically hydrostatic (with a gradient $\rho g$) or hydrostatic perpendicular to the velocity (with a gradient $\rho g \cos \beta$). The derivation in the textbook of Greve and Blatter (2009) already starts from a horizontal bed and horizontal velocity, so that it is not surprising that they arrive at vertically hydrostatic pressure. In the meantime, I realized that the old textbook of Hutter (1983) already contains the version with an inclined bed and hydrostatic pressure normal to the bed, which hopefully helps to support my version. If the ice surface is parallel to the bed, this version is also consistent with the parallel sided slab considered in the textbook of Greve and Blatter (2009). The reviewer points out the the parallel sided slab cannot be mixed with the SIA, but I would say that the two approximations must even agree in this specific situation. As requested below, I also provided a sketch of the proof why the pressure must be hydrostatic normal to the velocity. **In the paper, I tried to make it a bit more convincing for those readers who have only seen the version with the horizontal bed so far (lines 93–98).**

*Further, I would expect a clear proof of the advantage of running SIA on – how I conclude – 2 orders of magnitude above the with theory consistent mesh resolutions and quantify the errors introduced by steep slopes/high accumulation/fast sliding to comparison of either observed results or results obtained with to the task suitable approximations to or the Stokes equations. This not necessarily has to be done on domains as big as whole parts of the Alps.*

Here I disagree to the 2 orders of magnitude. I tried to explain in my first response that the grid spacing should not be confused with the horizontal scale of the topography. A deep, 25 m narrow canyon with 600 m ice thickness would, of course, make no sense with the SIA. **The new Sect. 6 shows a quite steep valley with a thickness-to-width ratio of about 0.25 as some kind of worst case for the considered topography.** The ratio of 0.25 (or perhaps 0.5 if we use half of the width as the horizontal scale) is definitely not close to zero as theoretically required for the SIA, but for sure not 2 orders of magnitude too high. Admittedly, the results for $\delta x = 25$ m, 50 m and 100 m do not differ much and the physical error of the SIA is probably already higher than the numerical error. However, the topography does not only consist of such narrow valleys. So I do not follow the argument that high resolutions are useless in combination with the SIA.

*The second concern is linked to the main motivation given by the author to provide a computationally fast method to the community in order to compete with machine learning (ML) algorithms. My deviating opinion on this competition aside (see in specific comments), I see the following issues concerning the description of model performance and reproducibility, namely*

*- the proprietary package needed to run the implementation (MATLAB) prohibiting reproduction of the results for readers without a valid license*

*- the lack of detailed description on applied parallel computation paradigms (if any), like OpenMP (threading and SIMD) or even distributed memory, like MPI. Can the setup perhaps even utilize accelerators?*

*- the detailed justification on the choice of the methods (one direct and one Krylov) used to solve the sparse matrix problem*

If I had realized before that MATLAB is such a huge problem, I would have added a 0.x version in Python with the direct solver and the CG solver (without preconditioning) from SciPy. Anyway, the Python version is available now. I finally ended up with a short C++ code for the PCG scheme and the preconditioner (about 100 lines in total), which has to be included as a precompiled .so (Linux, MacOS) or .dll (Windows). While the C++ code must be compiled by the user for Linux, a precompiled .dll for Windows is already included in the repository. The Python version was tested extensively with several of the benchmark scenarios and the numerical efficiency is surprisingly good (even 1.5 times as fast as the MATLAB version).

And as the computational efficiency is so much higher than in other models, I did not think about parallel computing so far. **I added a short note (lines 336–338).** As a first estimate, the choice of the preconditioner may change if there is a large number of cores available since the incomplete Cholesky decomposition is the only part that is inherently sequential.

Concerning the options of a direct solver and a PCG solver, I feel no need to justify this choice. As stated in the text, the direct solver is just for testing and PCG with incomplete Cholesky came from experience with other diffusion-like problems. Users should feel free to include alternative solvers.

**Specific comments**

*Line 2: "This paper aims at keeping classical numerics competitive in this field." This is merely a comment conveying my divergent opinion: I do not think that the competition (if there is, as I rather see them as complementary) between machine learning (ML) algorithms and process based models (classical in terms used here) is about speed. In my opinion, it is about accuracy and physical completeness. ML reproduces what it has been trained with, and it does this very fast and efficient. The particular reason why these techniques (or at least in my opinion should be) utilized is exactly to speed things up. On the other hand, ML is restricted to a given setup (fixed set of geometry and/or processes) defined by the training set. These surrogate models need new training (which tends to be computationally intensive, so no free lunch also here) to introduce new physics, often even if moving to another topography. That – in my opinion – is where the strength of process models lies – they easier can introduce improved physics and they are more generally applicable. I, though, think it is futile to try to compete in terms of speed with a ML implementation using the same setup in the training set as in a process-based model (of which complexity it ever may be), in particular, if the training phase of the ML is left out of the comparison.*

I share this point of view , but my knowledge about AI-based approaches is not sufficient. In particular, I cannot assess the reliability of the "physics-informed deep learning" version published in 2023. But as a spectator, I see that the AI-based approach has motivated several researchers to consider numerical modeling of ice flow. To me, it seems that the huge numerical effort (I accept that the reviewer may have a different opinion what huge means here) of classical numerical models kept researchers off from this field for several years.

*Line 5: "... a lightweight implementation of the new scheme in MATLAB." This relates to the second major point before. In my opinion, it would have been better to wait for the Python implementation to be ready before publishing. At this point in time, the reader needs a licensed software (MATLAB) to reproduce results and is left without any insight on the performance of the Python implementation to come. To palliate the first problem, perhaps the author can test with the open source software Octave and report to what extend this can be used in replacement to MATLAB to run your current version of the code?*

Thanks for the suggestion and sorry for not making clear that the problem with the efficiency of the Python version will be solved in a reasonable time. With this precompiled solver, the Python version it is even about 1.5 times as fast as the MATLAB version. And since it is already about 2–3 orders of magnitude faster than an explicit scheme, I will not evaluate the parallel-computing performance at the moment.

*Line 16: "Three-dimensional simulations of the Stokes equations with a free surface (as implemented, e.g., in the model Elmer/Ice) and the shallow ice approximation (SIA) are end-members in the hierarchy of ice-flow models." May I in this connection point out that there are existing Stokes simulations of the Western part of the Alps (in fact, including Black Forest, which was not part of the investigation but the domain) spanning several thousand years around the LGM (Cohen et al, 2018 and 2023).*

I remember discussing with Denis Cohen about his simulations of the Rhine glacier long ago and that he told that full Stokes simulations are not much more expensive than PISM simulations. **I added the older reference (lines 34–37).**

*Line 17: "The SIA considers vertically averaged velocities, assumes hydrostatic pressure, and neglects all stresses arising from horizontal shearing." To my knowledge, SIA defines horizontal velocities as a function of the vertical coordinate (Greve and Blatter, 2009), but only its force- balance considers vertically integrated stresses (expressed in terms of integrals of the vertically varying horizontal velocity field along columns). So, the fluxes, but not the velocities are vertically integrated variables. Could it be that in the presented application the fact that one would have to perform a quadrature of the force balance in each column is circumvented by reducing the solution to the mass-balance and the introduction of the pre-factors $f_d$ and $f_s$ therein? If so, I would ask to explain what assumptions the replacement of the vertical integrals by – what occurs to me – fixed pre-factors are implicitly introduced by this procedure (e.g. on vertical distributions of enhancement factors, temperatures).*

The "vertically averaged velocities" were a remnant of shallow-water or Savage–Hutter equations. Of course, the velocity is not an independent variable in the SIA, but defined by thickness and gradient of the ice surface. **I adjusted the wording (lines 20–21).**

The vertical profile of the horizontal velocity is the integral given in Eq. A3. The fluxes are then obtained by integrating once more (Eq. A4) and the depth-averaged velocity is just flux divided by thickness. **I extended the derivations in Appendices A and B bit by replacing the proportionalities with the respective factors.**

As soon as the mechanical properties of the ice become depth-dependent, the integrals have to be evaluated with a depth-dependent deformation parameter $A$. Without the slope correction, this was elaborated for the example of the the parallel-sided slab with a linear temperature profile in the textbook of Greve and Blatter (2009).

*Line 18: "In between, there are two-dimensional approaches that account for these stresses, ... ." Even Stokes can be solved in two dimensions (flow line). I would suggest to write depth-integrated instead.*

Yes, of course! I did not think about flow-line approaches because the mass balance perpendicular to the flow line is crucial in the examples I have seen so far. **I adjusted the wording (line 24).**

*Line 26: "The still limited performance of two-dimensional models arises from a combination of assuming hydrostatic pressure and the explicit time-stepping scheme implemented in almost all contemporary models (Bueler, 2023)." Like before, I would suggest using depth-averaged or integrated models, instead. Further I would ask to explain what performance is referred to? Pure computational performance or beyond that also in terms of accuracy and stability? In view of the latter (which links to my primary point of criticism), I would drop the "still", as to me it implies that the instabilities/inaccuracies described in this manuscript are something that could be overcome. In my opinion, they are built-in, as models based on hydrostatic approximation are not capable to resolve horizontal gradients in stresses that act on scales smaller than a few multiples of the ice thickness and assume small slopes (Hindmarsh, 2004).*

Performance is numerical performance here for me, but this performance is dominated by the instability of the usual schemes. So not accuracy in the context of narrow valleys, steep slopes or strong sliding etc. **I adjusted the wording (line 38).**

The other point reflects our main disagreement. I agree that the hydrostatic approximation cannot resolve horizontal gradients in stresses that act on scales smaller than a few multiples of the ice thickness. However, as explained in the general part, I strongly disagree to the opinion that the numerical issues are an inherent property of the SIA and thus cannot be fixed by an improved numerical scheme.

*Line 62, eq. (3) and line 68 and eq. (5): "The factor involving the cosine of the slope angle $\beta$ of the ice surface with ... is a simple correction for steep slopes." I will come back to this later also in the appendix, but to me it appears that introducing this slope correction violates a principal assumption of the SIA that the surface normal in lowest order points exactly into the vertical direction which by shallowness also would demand that $\cos\beta \approx 1 \rightarrow \beta \approx 0$. In the other extreme, for $\beta \rightarrow \pi/2$ the 5th and 8th power of $\cos\beta$ in (9) will result in a vastly vanishing pseudo diffusivity and (8) basically reduces to the geodetic mass balance. In other words, mass transport shuts down on steep slopes. That in my view is artificial and covers the fact that the hydrostatic pressure approximation is not valid on slopes significantly deviating from zero (Hindmarsh, 2004) – see point one in major critics.*

The argument with $\beta \rightarrow \pi/2$ is technically correct for a given vertical thickness. Here it happens because the thickness perpendicular to the bed goes to zero. This points towards a major disagreement (see general points and below): The reviewer says that the vertical thickness is responsible for the flow velocity, while I say that it is the thickness perpendicular to the surface. For discussing the effect of the slope correction, however, given vertical thickness is not the right point of view. It would rather be a given flux since the ice from the upper parts has to be transported. At given flux, the vertical thickness is then proportional to $(\cos\beta)^{-8/5}$, where $(\cos\beta)^{-3/5}$ comes from $\sin\beta$ instead of $\tan\beta$ and $(\cos\beta)^{-1}$ from converting vertical to surface-normal thickness. So there is no technical problem with the slope correction, and the question whether the SIA can be written in a rotated coordinate system is discussed below.

*Line 67: "The factor $f_d$ mainly depends on temperature in reality." This factor should also contain some contributions from the vertical integral of the force balance, including the temperature dependence and enhancement factors. As demanded before, I think for completeness, the exact composition of $f_d$ as well as $f_s$ leading to the constant values reported in the text should be written out and explained – if not in the main part, at least in the Appendix. Further, as this seems to be the only mentioning of temperature, which, by the Arrhenius factor (Greve and Blatter, 2009) has a strong influence on the ice viscosity and hence the flow: I would kindly ask how the model – if at all – treats temperature variations in the ice, in particular as I would think that during a glaciation of the Alps the thermal ice conditions significantly vary, both, spatially and in time? Would the presented model be in principle capable of including advective temperature transport?*

**I replaced the proportionalities in Appendices A and B by the full expressions for the case that $A$ and $\rho$ are vertically constant.** In this case, the exponential Arrhenius factor would just occur as a factor in $f_d$. Otherwise, the integrals have to be evaluated with depth-dependent parameter values.

Advective heat transport could be included easily as long as we are not interested in the vertical temperature profile because we could directly derive the energy flux from the ice flux. Depending on the scheme, we would just have to reduce $\delta t$ not to exceed the CFL criterion. And I would add a dispersion term to take into account the vertical velocity profile. If we include the vertical temperature profile, it would become technically more complicated and I am not sure whether I would do this with such a simple model.

*Line 70: "The correction is explained in the derivation of Eq. (3) in Appendix A. It should, however, be mentioned that it is less elaborate than the correction for rapid mass movements developed by Savage and Hutter (1989) and only exact if the ice surface is parallel to the bed. In particular, it does not capture the effect of steep walls in a valley." I discuss this in detail later where the method is explained in the Appendix. As mentioned before, I think the cosine in (3) is in violation of the basic assumptions behind SIA. I also do not see the parallel slab problem of a creeping shear thinning fluid in a direct connection to the Savage-Hutter theory, which is an expansion of the depth-averaged Navier–Stokes equations (i.e., including acceleration terms) for yielding granular flows, neglecting terms of order $O(\epsilon^{3/2})$ of the aspect ratio $\epsilon$, and in contrary to the SIA, usually already defined in a locally rotated coordinate system (e.g., Zwinger et al., 2003).*

I explained briefly in the general part and below why I am sure that it is correct and in fact consistent with the parallel slab solution. **I adjusted the wording here and added the references Hutter (1983) for the general framework and Greve and Blatter (2009) for the parallel sided slab (lines 93–98).**

Line 80: *"It should, however, be emphasized that modeling sliding along the bed is still one of the major challenges in this field. It is even questionable whether this process can be described well by the SIA or whether lateral stress components have to be taken into account." Please, refer to literature where these topics are addressed. For the latter: It is established (see literature cited in the major points section) that SIA is unsuited for fast sliding conditions.*

**I rephrased the sentence and included the references (lines 106–108).**

Line 89: *"Formally, it looks like a nonlinear diffusion equation, although it is mathematically not a diffusion equation due to the occurrence of $\nabla s$ in $D$. Can you please elaborate: If not a diffusion equation, what is it then?*

It is still a parabolic PDE in the part of the domain where $\nabla s \neq 0$ because its linearized form is parabolic. Mathematical studies on nonlinear diffusion, however, typically do not allow $D$ depending on $\nabla s$. The reason is that the linearized form has a different diffusivity of $\tilde{D} = \frac{\partial}{\partial |\nabla s|}(D|\nabla s|)$, which would be $\tilde{D} = 3D$ here. Then all properties (diffusion of small disturbances, Fourier criterion, ...) depend on $\tilde{D}$ instead of $D$. I want it to be mathematically correct, but not to go into detail. **So I rephrased the statement (lines 120–121).**

Line 120: *"The upstream scheme ensures $D = 0$ in this situation and thus avoids a permanent systematic increase in ice volume." What about the considerations on the second term of the r.h.s. of (8) at the contact line in order to avoid negative heights? There is often negative net mass balance, $r$, at the front of the glacier. Is it masked out? Generally speaking, to stay consistent in mass balance one should solve the mass balance or in higher order models the kinematic boundary condition together with the inequality for positive or zero flowheights (see, e.g., Gagliardini et al. 2013, section 6.5), which in my opinion should be applicable to the scheme applied in this manuscript.*

There is no minimum thickness $h_{\min}$ implemented here, but it could be. And the workflow is:

1. Compute $D$

2. Compute $r$ and update the ice surface

3. Solve the ice flow equation

4. Remove negative thickness values

Steps 2 and 3 could be interchanged because the gradient in $r$ is typically small. If the thickness is negative and lower than $r\delta t$ (so more ice lost than melting), it is considered an error in the mass balance. At the front, however, this does not happen.

*Line 128: "Therefore, a dynamic smoothing of the slope of the ice surface in the diffusivity $D(h, |\nabla s|)$ is the central novel idea in MinSIA. The idea was originally motivated by Airy's linear theory of ocean waves (gravity waves in an inviscid fluid with a free surface). According to this theory, oscillations in pressure and particle velocity decrease exponentially with depth below the surface. The respective depth of penetration is proportional to the wavelength. Based on this result, MinSIA uses a smoothed version of $|\nabla s|$ with the length scale of smoothing being proportional to the ice thickness h." I am not able to see a connection from a theory based on gravity waves in Newtonian fluids to the mechanical behaviour of a (strongly non-linear and highly viscous) shear thinning free-surface thin-film Stokes problem. My theory: this smoothing algorithm works because it re-introduces the basic principle of thin-film approximations to resolve horizontal gradients with respect to a large aspect ratio (i.e., over several ice-thicknesses), even if run on mesh resolutions that apparently violate this assumption. To probe that theory, I suggest to run the simulation on a to the SIA appropriate mesh resolution ($> 1000$ m range) without smoothing and report on instabilities of such a run. I would also ask to include a clear motivation to opt for such small horizontal resolutions of 25 m–100 m in connection with SIA. Thinking of the main motivation presented in the text, a fast solution, I would even say that increasing the resolution should work exactly in this direction as the problem size gets smaller. This links to the first in the main point of critics.*

**I removed the statement about Airy's theory and extended the numerical reasoning a bit (lines 166–174).** Further details are given in the response to the respective comment by the first reviewer.

Technically, I agree to the theory proposed by the reviewer. If we use a sufficiently large grid spacing, smoothing vanishes and even explicit schemes become stable. So there is no need to probe the theory technically. As discussed in the general part, our disagreement that the numerical issues are not related to the physical deficiencies of the the SIA from may point of view. In my opinion, the SIA does not become increasingly wrong just because we use finer grids and a coarse resolution that does not capture the valleys would not be better.

*Line 163: "The preconditioned conjugate gradients (PCG) method is used for solving the linear equation system. An incomplete Cholesky factorization is used as a preconditioner, whereby the version that compensates dropped nondiagonal elements at the respective diagonal elements turned out to be particularly suitable. A direct solver is also implemented for testing." As mentioned in the second point of major critics. I would ask for a more detailed explanation of this particular choice. What method was used for the direct solver? Were these methods utilizing specific features of the hardware (shared-memory or distributed memory parallelism, vector units, accelerators, if present)? What can the reader expect if this model is run on other, more modern platforms than the CPU that was reported in the text? Are there special license conditions to utilize the mentioned methods in MATLAB?*

*Line 187: "As a first test, the accuracy of the solutions for different values of the smoothing factor $f$ with respect to the reference solution ($f = 0.25$, $\delta t = \frac{1}{64}$ yr) is measured." In relation to the major point of critics, I do not think that comparison to a result obtained with the method is able to show the accuracy of the method itself. To me, it merely shows the sensitivity of the method to a change in numerical parameters and resolutions.*

*Line 209: "Since even explicit schemes are stable for sufficiently small time increments, the occurrence of the staircase oscillation must be related to both the smoothing factor $f$ and the time increment $\delta t$." I would like to see a proof on the statement of stability for small timesteps. Either by citing literature or running the model (at least for some time) with no smoothing on the 25 m resolution. What I think, is that the oscillations are a reaction of the SIA to significant undulations with a length-scale way below the ice-thickness. Consequently, I would conclude that the oscillations are natural to the SIA (if run on resolutions that violate the shallowness assumption) and damped out by the smoothing, similar to as they would by running the SIA without smoothing on coarser meshes (which I already asked you to test) that resolve horizontal scales above the typical ice-thicknesses.*

The linear solver is not an essential part of the scheme and it should be easy to replace it by any other solver if desired. So I feel not need to justify the specific choice. I am even not sure which direct solver MATLAB uses by default, but the direct solver is only interesting for small test examples since it requires much memory. The preconditioner was found by performing experiments with several different preconditioners and settings, but already knowing that incomplete Cholesky factorizations are not too bad for problems of this type. **I added a short statement about the absence of parallel computing features beyond the default vectorization of MATLAB (lines 336–338).** In principle, everything except for the steps related to preconditioning could be parallelized. So there might be a scale at which the preconditioner becomes the bottleneck, so that another preconditioner might become better.

I thought that it was already clear that the deviation between the solution with $f = 0.25$ and that without smoothing (not computed) is small. **Following the suggestion, I added some results from a shorter simulation (from $t = 2900$ yr to $t = 3000$ yr with $\delta t = \frac{1}{4096}$ yr) to illustrate that the difference between smooth-free ($f = 0$) and reference ($f = 0.25$) is clearly smaller than the difference between $f = 0.25$ and $f = 0.5$ (lines 257–264 and new Fig. 4)**. Nevertheless, accuracy is only considered within the framework of the SIA and not with regard to a full Stokes simulation.

The point that the oscillations arise from undulations with a length-scale way below the ice-thickness is correct (the oscillations with the worst impact are on the scale of $\delta x$), but the rest of the statement reflects our main disagreement. The SIA is practically a diffusion equation (the fact that the linearized diffusivity is $3D$ instead of $D$ does not matter here), which means that these undulations do not result in growing oscillations for an exact solution. Exact solutions of diffusion-type equations smooth such undulations without overshooting. The exact solution dampens such undulations even faster if the wavelength is shorter. The conversion of surface undulations into oscillations is fully a matter of the numerical scheme. This is confirmed by the simulation without smoothing and $\delta t = 1/4096$ yr reported in the previous point.

Line 267: "The simulations over the entire time span with $\delta t = 0.25$ yr took $35400$ s for $\delta x = 25$ m, $6740$ s for $\delta x = 50$ m, and $1290$ s for $\delta x = 100$ m on an Intel Core i5-7600 CPU (3.50 GHz) from 2018." This, to me, seems to be the only information on how some sort of performance evaluation has been done. In my view, this is not enough such that the reader can get a clear picture what to expect if running the model themselves. I would ask to specify the utilization of the CPU (single core or multiple cores, as there seem to be 4 included in the reported model). If multiple cores have been utilized, please report on what parallel paradigm was deployed (shared or distributed memory). How much memory was available in the test platform and how was the saturation of the available memory at the runtime? This links to the second point of my major critics.

**I added some more information about the number of cores (lines 336–339).** Memory usage is not worth mentioning – about 1 GB for the Python version on the $2400 \times 2400$ lattice. It scales linearly with the number of cells for the iterative solver. It should be immediately clear that it is much faster than Elmer/Ice and PISM, but not a comprehensive model including heat transport etc. and with the limitations of the SIA. From my experience, it is also much faster than the implementation of the iSOSIA in the respective erosion model. And a comparison with the AI-based IGM seems to be extremely difficult to me at the moment (usage of GPU computing, time for training, . . . ).

Line 288: "Figure 8 illustrates the weakness of the scheme at steep slopes for $\delta x = 25$ m and $\delta t = 0.25$ yr." I would rather say it reveals the inapplicability of the approximation than a weakness in the numerical scheme (see major point of critics).

Definitely not at this occasion. As written in the text, this applies to steep slopes with rather small ice thickness. Here the shallow-ice or slab approximation itself is ok. It is just that advection dominates over diffusion at these points and that the scheme is not better than any other scheme then.

Line 305: "At present, a MATLAB implementation of MinSIA is available under the GNU General Public License. A Python implementation is under development." As mentioned under the major points and before in this section – despite the setup being shared with an open license – the proprietary nature of the software needed to run it imposes a big hurdle for people to reproduce these findings. This all would have been no problem if the publication would have been postponed to the point in time when the announced Python version would have been available.

I hope that is point was addressed sufficiently in the response to an earlier comment. In this specific case, I could imagine that the reviewer would indeed have tested it without smoothing and a small $\delta t$. But I honestly do not know many reviewers who would have spent this work.

Line 334: "A 2-element array corresponds to $r_{\max} = \infty$ and a scalar value additionally to $g_- = g_+$." I am not able to understand the meaning behind this sentence. Can you please elaborate? Do you allow for infinitely large accumulation rates?

It just means that the third entry of the array can be omitted if if unlimited ice production is desired and that the second entry can be omitted if the ablation gradient shall be the same as accumulation gradient. May be useful or not, but it is implemented like this and should in principle be clear from the text.

*Line 390: "Let us, for simplicity, assume that the ice surface is parallel to the bed with a slope angle $\beta$ and that the $z$-coordinate is perpendicular to the surface with $z = 0$ at the bed." I think this is the culprit of the discrepancy about the bed-slope correction I raised in the main issues. The parallel slab example (see Greve and Blatter, 2009), which seems to be adapted here, in my view, cannot be intermixed with the derivation of the SIA. The basic assumption behind the SIA is a directly into the vertical direction pointing surface normal (Greve and Blatter, 2009). The resulting full alignment with the negative direction of the gravity is a necessary condition for the zero Cauchy stress vector at the surface leading to identically vanishing stress-components and pressure, which constitutes another necessary condition for the vertical integration of the shear stresses (Greve and Blatter, 2009) in SIA. If the integration in the vertical direction is done from bedrock, $b$, to the free surface $h$, one obtains the shear stress at the bedrock as $\tau = \rho \rho g h \nabla s$, where $h = s - b$, i.e., the total ice-depth in vertical direction. Thus, in SIA the shear stress is imposed by the hydrostatic pressure gradient induced by the horizontal change in surface elevation (with no influence of the bedrock slope at all), whereas in the parallel slab, i.e., equation (A1), the shear stress is the result of the downslope weight component of a in the rotated reference frame bed-parallel surface with in this rotated reference frame identically vanishing gradients of the free surface. As I see it, the author is setting (A1) identical to the bed-shear stress as defined in SIA, which I claim violates the basic assumptions of the SIA. If the author disagrees, I would ask to include a detailed derivation of the SIA (starting from the Stokes equations) in a rotated reference frame and show that the derived equations still comply with the lowest order approximation, even for large bedrock angles, as they occur in the studied topographies of the Alps and even the Black Forest.*

As written above, I am quite sure that my version is correct and is consistent with the version from the textbook of Hutter (1983). For simplicity, I write the requested proof only for the linear Stokes equation with constant viscosity,

$$-\nabla p + \rho \vec{g} + \eta \Delta \vec{v} \; = \; 0,$$

and the same direction of the velocity everywhere,

$$\vec{v} \; = \; f(x, y, z)\vec{e},$$

with a (constant) unit vector $\vec{e}$. Then

$$\Delta v \; = \; (\Delta f)\vec{e}.$$

Define a normal vector $\vec{n}$ perpendicular to the velocity, so $\vec{e} \cdot \vec{n} = 0$ and obtain

$$\nabla p \cdot \vec{n} = \rho(\vec{g} \cdot \vec{n}),$$

which means that the pressure gradient perpendicular to the velocity is the component of gravity perpendicular to the velocity.

For completeness, the hydrostatic pressure at the bed (valid for Stokes and Navier–Stokes) in Cartesian coordinates (see, e.g., Hergarten 2024, doi 10.5194/gmd-17-781-2024) is

$$p_b = \frac{\rho g h}{1 + \nabla s \cdot \nabla b}.$$

The SIA version for a horizontal bed ($p_b = \rho g h$) is recovered for $\nabla b = 0$ and the slab version (my correction terms) for $\nabla b = \nabla s$, which leads to

$$p_b = \rho g h \cos^2 \beta.$$

**Typos and technical corrections**

*Line 245: PGC $\to$ PCG*                    **Fixed, thanks (line 301)!**

*Line 281: ... which is also is still stronger than ...*                    **Fixed, thanks (line 352)!**

*Line 396: $\tau(z)^n$ suggestion to change to $\tau^n(z)$*                    I do not like this notation, but we can afford to write $(\tau(z))^n$ **(line 521, Eq. A2)**.

*Line 409: suggestion: first term $\to$ first multiplier on the right-hand-side*                    I had ho change the wording anyway because I included the factor of proportionality.

**Technical comment (Katja Gänger)**

*Figure 1 contains satellite, drone, or airborne images. If you are not the originator of the images, then appropriate credit or copyright must be given. If applicable, please add the necessary details to the figure or the figure caption. Please make sure that the figure or caption contains the appropriate image credit as this is the responsibility of the authors. Further details are available at: https://www.geoscientific-model-development.net/submission.html#mapsaerials*

Figure 1 is a self-made plot of the referenced elevation model combined with simulation results.